# Shared structural features of Miro binding control mitochondrial homeostasis

Christian Covill-Cooke (ID)[1 ✉], Brian Kwizera[1], Guillermo López-Doménech (ID)[2], Caleb OD Thompson[2], Ngaam J Cheung[1], Ema Cerezo[1], Martin Peterka[3], Josef T Kittler (ID)[2] & Benoît Kornmann (ID)[1 ✉]

## Abstract

Miro proteins are universally conserved mitochondrial calcium-binding GTPases that regulate a multitude of mitochondrial processes, including transport, clearance, and lipid trafficking. The exact role of Miro in these functions is unclear but involves binding to a variety of client proteins. How this binding is operated at the molecular level and whether and how it is important for mitochondrial health, however, remains unknown. Here, we show that known Miro interactors—namely, CENPF, Trak, and MYO19—all use a similar short motif to bind the same structural element: a highly conserved hydrophobic pocket in the first calcium-binding domain of Miro. Using these Miro-binding motifs, we identified direct interactors de novo, including MTFR1/2/1L, the lipid transporters Mdm34 and VPS13D, and the ubiquitin E3-ligase Parkin. Given the shared binding mechanism of these functionally diverse clients and its conservation across eukaryotes, we propose that Miro is a universal mitochondrial adaptor coordinating mitochondrial health.

**Keywords** AlphaFold; ERMES; Lipid Transport; Mitophagy; Organelle Transport
**Subject Category** Organelles

## Introduction

Mitochondrial function is tightly modulated by homeostatic mechanisms affecting their position, morphology, turnover, and protein and lipid composition. One highly conserved protein family appears central for mitochondrial function, the calcium ($Ca^{2+}$)-binding Miro GTPases. Miro binds and often recruits to mitochondria an array of client proteins that are effectors of all of the above processes. These include cytoskeletal adaptors (Trak (Fransson et al, 2006; Glater et al, 2006; MacAskill et al, 2009), CENPF (Kanfer et al, 2015; Peterka and Kornmann, 2019; Kanfer et al, 2017) and MYO19 (Oeding et al, 2018; López-Doménech et al,

2018; Bocanegra et al, 2020)), lipid transport contact-site factors (the ER–Mitochondria Encounter Structure, ERMES in yeast (Kornmann et al, 2011; Stroud et al, 2011) and VPS13D in metazoans (Guillén-Samander et al, 2021)) and the mitochondrial quality control E3-ubiquitin ligase Parkin (López-Doménech et al, 2021; Safiulina et al, 2019; Shlevkov et al, 2016) which degrades Miro upon mitochondria-specific autophagy induction (Wang et al, 2011), the failure of which is a hallmark of both idiopathic and familial Parkinson's disease (PD) (Hsieh et al, 2016). The functional diversity of Miro clients raises important questions: how does Miro accommodate binding to so many clients? Do they bind simultaneously as (a) large complex(es) or successively through competitive processes? And what significance does this have for the coordination of organelle homeostasis? Miro comprises two GTPase domains (GTPase1 and 2) flanking two $Ca^{2+}$-binding EF hand with LM helices (ELM1 and 2) (Fig. 1A) (Fransson et al, 2003; Klosowiak et al, 2013; Smith et al, 2020). Structural information has been gathered on all these domains (Klosowiak et al, 2013; Smith et al, 2020), but how Miro binds its partners at the structural level is unexplored.

## Results

### Identification of a hydrophobic client-binding pocket in Miro

To address these questions, we sought to understand how Miro binds its clients. We focused on CENPF on account of its well-defined Miro-binding domain; namely, 42 amino acids within CENPF C-terminus (CENPF-2977–3020) necessary and sufficient for direct Miro binding (Peterka and Kornmann, 2019; Kanfer et al, 2015). To address which Miro domain binds CENPF, MIRO1 truncations were generated and cloned into a yeast two-hybrid (Y2H) system with CENPF-2819–3114 as bait. We found that both the GTPase1 and ELM1 domains together were necessary and sufficient for CENPF binding (Figs. 1B and EV1). To understand the exact nature of binding, we used the AlphaFold2 multimer model with CENPF-2977–3020 and MIRO1 (Evans et al, 2021). AlphaFold2 predicted with high confidence that CENPF binds to MIRO1 at a highly conserved patch (Fig. 1C,D). CENPF-F2989—a

[1]Department of Biochemistry, University of Oxford, South Parks Road, Oxford OX1 3QU, UK. [2]Department of Neuroscience, Physiology and Pharmacology, University College London, Gower Street, London WC1E 6BT, UK. [3]Institute of Biochemistry, ETH Zurich, 8093 Zurich, Switzerland. ✉E-mail: christian.covill-cooke@bioch.ox.ac.uk; benoit.kornmann@bioch.ox.ac.uk

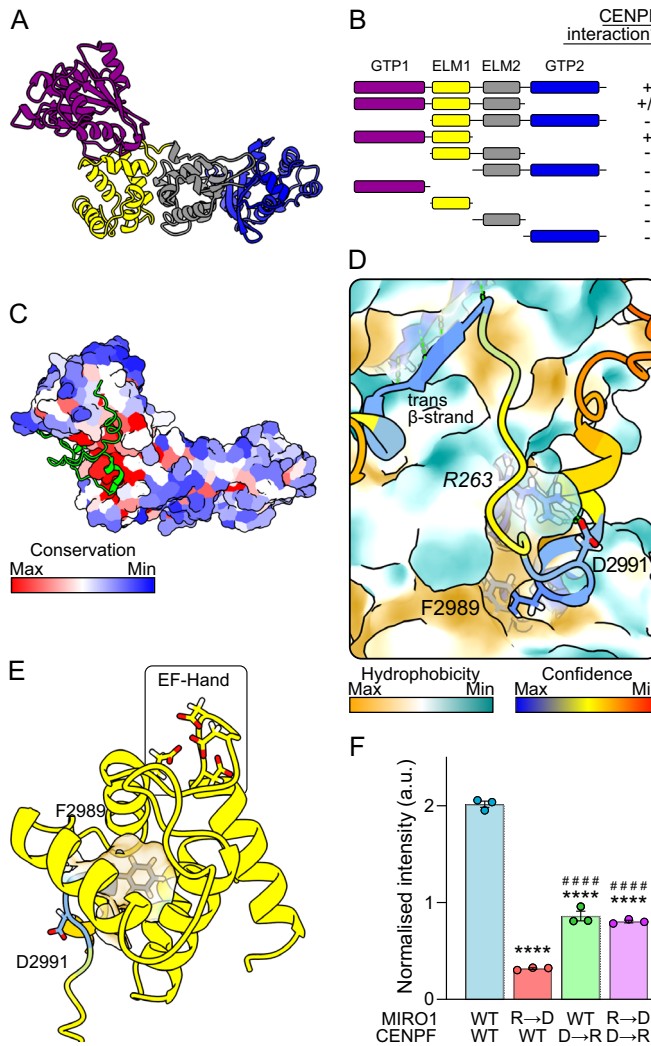

**Figure 1. CENPF binds to a conserved hydrophobic pocket in ELM1 of Miro.**

(A) AlphaFold2 predicted structure of human MIRO1 with domains color-coded: purple—GTPase1, yellow—ELM1, gray— ELM2 & blue—GTPase2. The C-terminal transmembrane domain has been removed. (B) Schematic showing which truncation constructs of human MIRO1 (prey) bind CENPF-2819-3114 (bait) in a yeast two-hybrid assay. + means an interaction was observed; - means no interaction was observed. (C) AlphaFold2 multimer prediction of CENPF-2977-3020 (shown in green) and human MIRO1. MIRO1 is color-coded according to amino acid conservation. (D) Zoom into the structure in (C). Color coding is by prediction confidence for cartoon and by hydrophobicity for MIRO1's surface. Italicized residues correspond to MIRO1 and non-italicized correspond to CENPF. (E) Structural features of the ELF pocket of MIRO1 (yellow) with inserted CENPF-F2989 (color-coded as in (D)). (F) Fluorescent yeast two-hybrid assay of wild-type MIRO1 or R263D mutant (R → D), and wild-type CENPF or D2991R mutant (D → R), n = three independent clones. Statistical significance was calculated by one-way ANOVA with a Tukey post hoc test. **** and #### denote P < 0.0001 in comparison to WT-MIRO1 + WT-CENPF and WT-MIRO1 + CENPF-D → R, respectively. Graph shows mean ± SEM. Source data are available online for this figure.

key phenylalanine residue previously shown to be essential for Miro binding in vitro and in vivo (Peterka and Kornmann, 2019)—inserts extensively into a hydrophobic pocket within MIRO1-ELM1 (Fig. 1D,E), opposite to the $Ca^{2+}$-binding EF hand, which we call ELM1-domain leucine- or phenylalanine-binding (ELF) pocket.

Alongside F2989, a conserved aspartate residue (D2991) is predicted to salt bridge with the conserved R263 on MIRO1 (Fig. 1D). In addition to these ELF-interacting features, a β-strand downstream of F2989 in CENPF (ILR; 3001–3003) makes an antiparallel β-sheet with a β-strand (IETCVE; 141–146) within MIRO1-GTPase1. To validate the AlphaFold2 prediction, we focused on the salt bridge formed by negatively charged CENPF-D2991 and positively charged MIRO1-R263. Using a quantitative fluorescence yeast two-hybrid assay (f-Y2H) as a readout for interaction, we found that mutating MIRO1-R263 to D reduces the interaction. This can be partially rescued by simultaneously mutating CENPF-D2991 to R resulting in a charge swap (Fig. 1F), thus confirming the interaction predicted by AlphaFold2. The CENPF-D2991R mutation alone had comparatively little effect on binding perhaps because MIRO1-R263 can establish compensating bonds with backbone oxygens (see below).

## ELF binding is shared with other Miro interactors

We next sought to understand whether other known interactors bind Miro with a similar configuration. Specific regions of the microtubule motor adaptor proteins, Trak1 and Trak2 (Milton in *Drosophila*), and of the myosin motor MYO19 have been shown to interact with Miro (Oeding et al, 2018; López-Doménech et al, 2018; Glater et al, 2006; Fransson et al, 2006) (residues 476–700 of mouse Trak2 (MacAskill et al, 2009) and 898–970 of human MYO19 (Oeding et al, 2018)). Therefore, we predicted the interaction of either Trak1, Trak2, or MYO19 Miro-binding domains with MIRO1 in AlphaFold2. All three proteins appear to interact via MIRO1's ELF pocket, with Trak1-L597, Trak2-L581, and MYO19-F948 inserted into the pocket (Figs. 2A,B and EV2A). Both Trak and MYO19 interacting residues show very high conservation (Fig. EV2B,C). Indeed, Milton and *Drosophila* Miro (dMiro) are also predicted to interact via the same mechanism (Fig. EV2A). 50 amino acid stretches around the pocket-interacting leucine/phenylalanine of mouse Trak1 and human MYO19 interacted with MIRO1 in a f-Y2H, with Trak1-L594A (mouse protein) and MYO19-F948A, point mutants abolishing the interaction (Fig. 2C,D), supporting the AlphaFold2 prediction.

Like for CENPF, MIRO1-R263 was predicted to salt bridge with either Trak1-D599 (D602 in human protein) or with oxygens in the backbone (Trak2). Accordingly, the MIRO1-R263D mutation reduced binding to Trak1, while Trak1-D599R had little effect (Fig. EV2D). In contrast to CENPF though, a charge swap did not rescue interaction, likely because the partial salt bridges made by MIRO1-R263 with Trak1's backbone are important. We could, however, validate MYO19-binding interface using a charge swap. MYO19-E954 was predicted to make a salt bridge with MIRO1-R261, instead of R263 (Fig. 2B). Yet, while MYO19-E954R mutation substantially reduced interaction with wild-type MIRO1, neither single mutant MIRO1 variants (R261D or R263D) significantly affected binding (Figs. 2E and EV2E). A double MIRO1-R261D-R263D mutant, however, impaired binding. The arginines in the dMiro crystal structure which correspond to human R261 and R263 are not resolved (Klosowiak et al, 2013), suggesting that flexibility in these residues' orientation accommodates various salt bridges. Charge swapping (i.e., expressing both MYO19-E954R and MIRO1-R261D-R263D) not only rescued, but significantly increased binding, (Fig. 2E), validating the predicted binding conformation.

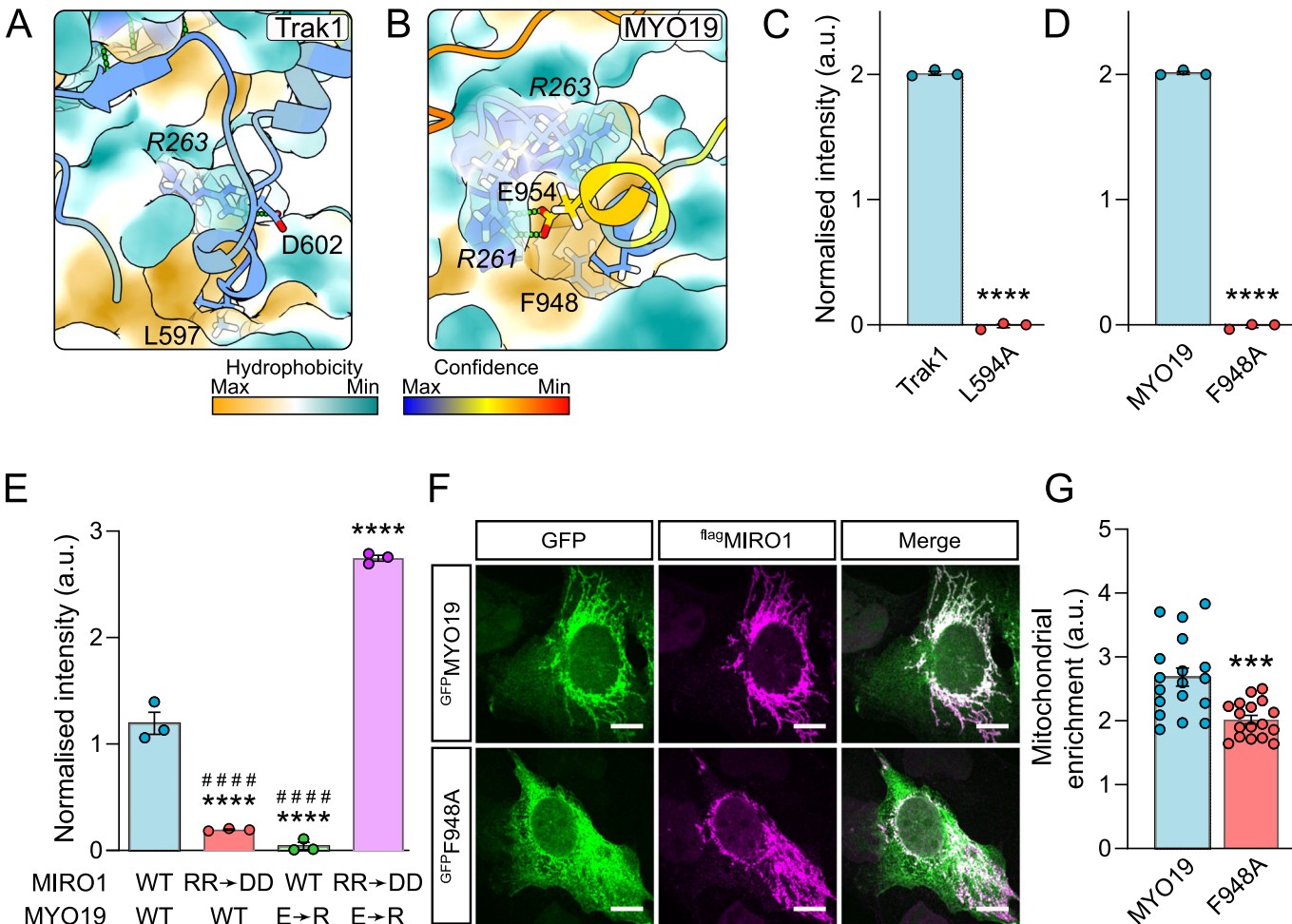

**Figure 2.   Trak1 and MYO19 bind to the ELF pocket of MIRO1.**

(A, B) AlphaFold2 multimer predictions of MIRO1 (surface) with Trak1 and MYO19 (colored as in Fig. 1D), respectively. Italicized residues correspond to MIRO1. (C, D) Fluorescent yeast two-hybrid assays of human MIRO1 with mouse Trak1-577-620 and human MYO19-919-970. (E) Fluorescent yeast two-hybrid assay of wild-type and R261R-R263D mutant (RR → DD) MIRO1 and wild-type or E954R (E → R) mutant MYO19-917-970. (F) Representative images of U2OS cells transfected with flagMIRO1 (magenta) and either wild-type or F948A GFPMYO19 (green). Scale bars represent 10 µm. (G) Quantification of the ratio of mean intensity of GFPMYO19 signal overlapping with flagMIRO1 over non-mitochondrial GFPMYO19 signal. N = 18 cells from three independent experiments. Data information: (C–E) N = three independent clones. (C, D, G) statistical significance was calculated by unpaired Student's t test. (E) statistical significance was calculated by one-way ANOVA with a Tukey post hoc test. *** is P < 0.001; **** is P < 0.0001 in comparison to WT conditions. #### in (E) denotes P < 0.0001 in comparison to MIRO1-R → D and MYO19-E → R. All graphs show mean ± SEM. Source data are available online for this figure.

To assess the relevance of these findings in vivo, we took advantage of the fact that the recruitment of MYO19 to mitochondria is partially dependent on Miro (López-Doménech et al, 2018). Overexpression of MIRO1 led to robust mitochondrial recruitment of MYO19, but not MYO19-F948A, which predominantly localized to the cytoplasm (Fig. 2F,G). A small amount of MYO19-F948A was recruited on mitochondria (Fig. 2F), likely due to the presence of features within the MYO19 C-terminus that allow mitochondrial localization independent of Miro (Bocanegra et al, 2020; Oeding et al, 2018). Consistent with the fact that Trak1 and Trak2 localize to mitochondria independently of Miro (López-Doménech et al, 2018), Trak1-L594A localized to mitochondria (Fig. EV2F). Thus, while we cannot exclude at this stage that additional molecular determinants play a role in the interactions, we find that the Trak proteins and MYO19 associate with Miro via a shared conserved binding pocket.

## A motif search identifies MTFR1/2/1L as Miro interactors

The identification of a shared mechanism of binding between CENPF, Trak1/2 and MYO19 to Miro raised the possibility that other proteins could interact with the Miro-ELF pocket. To explore this idea, we searched the mitochondrial proteome (MitoCarta3.0) (Rath et al, 2021) for a motif (FADI) based on the ELF-binding motif of CENPF. Of five candidates, we focused on MTFR2. MTFR2 is paralogous to MTFR1 and MTFR1L (Monticone et al, 2010): two mitochondrial proteins, which also have highly conserved potential Miro-binding motifs (MTFR1: FADV; MTFR2: FADI & MTFR1L: LADI) (Fig. EV3A). All three proteins were predicted by AlphaFold2 to bind to MIRO1 via the ELF pocket using either a phenylalanine (MTFR1 and MTFR2) or leucine

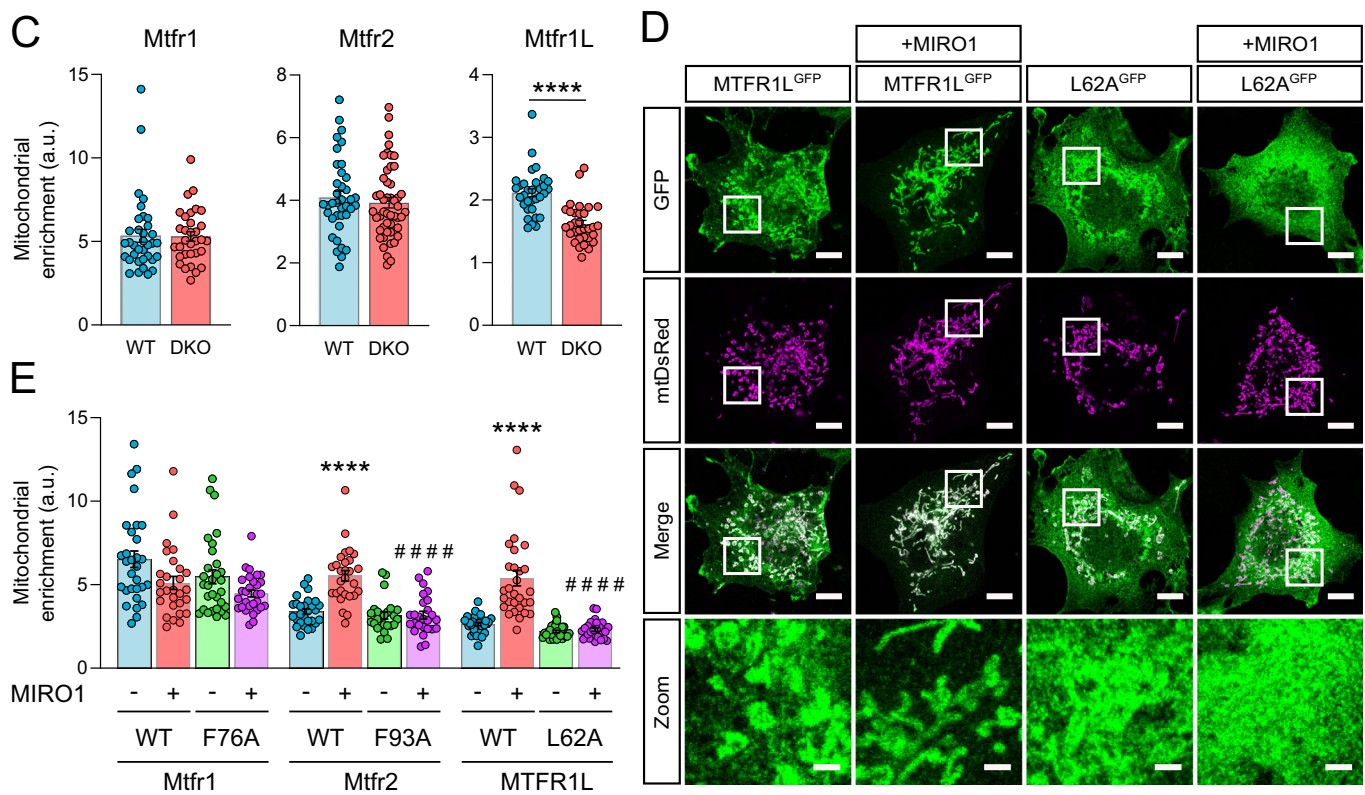

◄ **Figure 3.   MTFR1/2/1L as novel Miro interactors.**

(A) AlphaFold2 predictions of MTFR1, MTFR2, MTFR1L, and MIRO1 (colored as in Fig. 1D). (B) Representative images of myc-tagged mouse Mtfr1, Mtfr2 and Mtfr1l (green) in wild-type and Miro1/2 double knockout mouse embryonic fibroblasts. Mitochondria are stained with mtDsRed (magenta). (C) Quantification of mitochondrial localization of myc-tagged Mtfr1, Mtfr2 and Mtfr1l by calculating the ratio of mean intensity on the mitochondria over non-mitochondrial signal. $N = 32$–49 cells over five independent experiments. Statistical significance was calculated by unpaired Student's $t$ test. (D) Representative images of wild-type and L62A MTFR1L^GFP (green) in Cos7 cells transfected with and without ^myc MIRO1. Mitochondrial are stained with mtDsRed (magenta). (E) Quantification of mitochondrial localization of wild-type and point mutant Mtfr1, Mtfr2, and MTFR1L, both with and without MIRO1 overexpression, by calculating the ratio of mean intensity on and off the mitochondria. Statistical significance was calculated by one-way ANOVA with post hoc Tukey test. Data information: (B, D) scale bars represent 10 μm and 2 μm in zooms. **** is $P < 0.0001$ in comparison to WT conditions. #### is $P < 0.0001$ in comparison to WT + MIRO1. All data are shown as mean ± SEM. Source data are available online for this figure.

(MTFR1L) residue (Fig. 3A), and in all three cases, these interactions were confirmed using Y2Hs of full-length proteins (Fig. EV3B). Mutating the leucine or phenylalanine (Mtfr1-F76A, Mtfr2-F93A, mouse homologs, and MTFR1L-L62A, human homolog) reduced the interaction with MIRO1 (Fig. EV3B). Despite all proteins having an acidic residue near the Miro-binding motif, none were predicted to make a salt bridge with MIRO1. They were, however, predicted to contribute a trans-β-strand (MTFR1: ARL, 91–93; MTFR2: LRF, 91–93, MTFR1L: ARV, 77–79 in human sequences) to MIRO1-GTPase1 (Fig. 3A), like CENPF and Trak.

MTFR1, MTFR2, and MTFR1L localize to mitochondria (Tilokani et al, 2022; Monticone et al, 2010; Tonachini et al, 2004; Antonicka et al, 2020). To study if Miro was required for mitochondrial localization, mouse Mtfr1, Mtfr2 and Mtfr1l constructs were expressed in wild-type (WT) and Miro1/2 double knockout mouse embryonic fibroblasts (DKO MEFs). While Mtfr1 and Mtfr2 localized similarly to mitochondria in WT and DKO MEFs, and caused mitochondrial fragmentation as previously described, Mtfr1l failed to localize to mitochondria upon loss of Miro (Fig. 3B,C). To confirm the role of the Miro-binding motifs in vivo, we assessed the recruitment of Mtfr1-F76A, Mtfr2-F93A and MTFR1L-L62A mutants in Cos7 cells. In agreement with the DKO MEF microscopy data, Mtfr1 and Mtfr1-F76A localized to mitochondria, regardless of MIRO1 overexpression (Figs. 3E and EV3C). In contrast, MIRO1 overexpression caused increased recruitment of WT but not of the F93A Mtfr2 mutant (Figs. 3E and EV3D). Similarly, MTFR1L-L62A was not recruited to mitochondria by MIRO1 overexpression, in agreement with the DKO MEFs data (Fig. 3D,E). Therefore, all three MTFR proteins interact with Miro, two of which depend at least partially on Miro for mitochondrial localization. To sum up, a motif search identified novel clients which use the Miro-ELF pocket.

## Conservation of ELF pocket binding

The high conservation of the Miro-ELF pocket (Fig. 1C) and the varied interactors which bind it suggest conservation of this Miro-binding mechanism. We therefore set out to test if non-metazoan Miro orthologues also have this mechanism. Gem1 (Miro orthologue in *Saccharomyces cerevisiae*) is part of ERMES (Kornmann et al, 2011), a protein complex made up additionally of Mmm1, Mdm12, Mdm34 and Mdm10, that tethers the ER to mitochondria, allows efficient lipid transport between the two compartments, and is essential for tubular mitochondrial morphology (Kornmann et al, 2009; John Peter et al, 2022). How Gem1 interacts with other ERMES components is not currently known. By testing each of them, in AlphaFold2, we identified a disordered

loop in Mdm34 as interacting with Gem1 (Figs. 4A,B and EV4A). Importantly, this interaction was via a leucine residue (L248) inserting into the cognate ELF pocket of Gem1. An additional salt bridge is present but different from those found in metazoans, and involving Gem1-E242, a residue that is universally conserved, except in metazoans, highlighting divergent evolution. To test if L248 in Mdm34 is required for Gem1 interaction with ERMES, we took advantage of the fact that Gem1 colocalizes in puncta with Mdm34 at ER–mitochondria contacts (Kornmann et al, 2011) (Fig. 4C). Mutating Mdm34-L248 to alanine in the endogenous locus caused a complete dissociation of Gem1 from ERMES, resulting in a diffuse signal throughout mitochondria (Fig. 4C,D). Importantly, Mdm34-L248A formed foci and mitochondria remained tubular in this condition, indicating that ERMES function was not abolished. Therefore, ERMES binding to Miro's fungal orthologue is structurally similar to Miro clients in metazoans. This emphasizes the conservation of the ELF pocket across eukaryotes.

## Parkin and VPS13D bind the Miro-ELF pocket

Having identified a shared binding mechanism for several clients, we could assemble criteria to define binding motifs: (i) a conserved phenylalanine or leucine is required for pocket insertion; (ii) in mammals, at least, the F/L is often alongside an acidic residue; and (iii) the pocket-associating residues are in a conserved disordered loop. Using this knowledge, we set out to identify Miro-binding motifs in other proteins that associate with Miro, focusing on VPS13D and Parkin. VPS13D is a lipid transporter recently described as a Miro interactor which bridges the ER and mitochondria (Guillén-Samander et al, 2021), is essential in mammals (Blomen et al, 2015; Wang et al, 2015), and alleles of which cause recessive spinocerebellar ataxia (Seong et al, 2018; Gauthier et al, 2018). VPS13D's lipid-transporting function is homologous to that of ERMES, and yeast Vps13 and ERMES are partially functionally redundant (Lang et al, 2015; John Peter et al, 2017). Efforts to identify exactly where this interaction occurs on VPS13D and whether it is direct have not been fruitful but a so-called Vps13 adaptor binding (VAB) domain has been proposed (Guillén-Samander et al, 2021), partly by homology to yeast Vps13, which binds partners through this domain (Bean et al, 2018; John Peter et al, 2017).

A predicted structure of VPS13D, color-coded by conservation highlighted only two of the many unstructured loops as conserved (Fig. 5A): one comprising the phospho-FFAT motif required for associating to the ER via binding with VAP-A/B (Guillén-Samander et al, 2021), and the other, we term Miro-Binding Motif (MBM), adjacent to the VAB domain. This second loop contains a

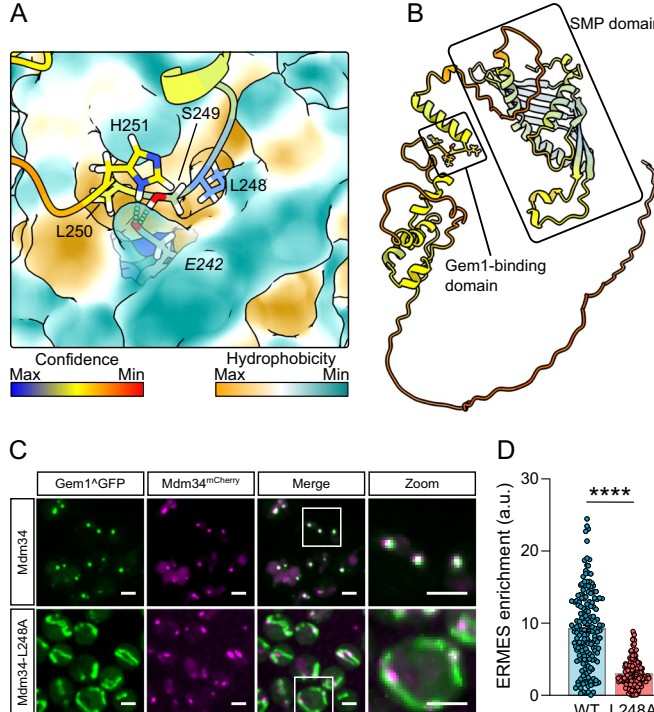

**Figure 4. Mdm34-L248 interacts with ELF pocket of Gem1.**

(A) Structural prediction of *Saccharomyces cerevisiae* Gem1 (surface) with Mdm34 (colored as in Fig. 1D). Italicized residue corresponds to Gem1. (B) AlphaFold2 predicted the structure of Mdm34 highlighting the lipid-transporting SMP-domain and Gem1-binding domain. (C) Representative images of internally GFP-tagged Gem1 (green) in wild-type and Mdm34-L248A budding yeast. Mdm34 was tagged with a C-terminal mCherry (magenta). Scale bars depict 2 µm. (D) Quantification of the extent of Gem1ˆGFP colocalizing with Mdm34-mCherry. $N = 168$ cells over two independent experiments. Statistical significance was calculated by unpaired Student's $t$ test. **** is $P < 0.0001$. Data are shown as mean ± SEM. Source data are available online for this figure.

conserved L2554 which AlphaFold2 predicted to insert into the ELF pocket (Fig. 5B). Alphafold2 also predicted that a trans-β-strand was constituted by residues 2449–2452. This long distance (~100 residues) between the trans-β-strand and the ELF-binding residue is unusual but the functional cooperativity between these two modules is made obvious by the fact that both are present/absent simultaneously in the same metazoan species (Fig. EV5A). To confirm the AlphaFold2 prediction, we used a mitochondrial recruitment assay. Overexpression of Miro caused significant recruitment of wild-type VPS13D to mitochondria (Guillén-Samander et al, 2021) (Fig. 5C,D). Remarkably, the recruitment of VPS13D-L2554A was severely blunted. We conclude that VPS13D binding to Miro-ELF pocket is a key part of its association with mitochondria.

Parkin rapidly ubiquitinates Miro during mitochondrial damage, as part of mitophagy (Wang et al, 2011). Miro over-expression increases Parkin recruitment to mitochondria irrespective of mitochondrial damage (Safiulina et al, 2019; López-Doménech et al, 2021; Shlevkov et al, 2016), but whether this is due to direct interaction is not known. An AlphaFold2 prediction of full-length Parkin with MIRO1 suggested that Parkin might bind to the ELF pocket using the conserved L119 (Figs. 5E,F and EV5B).

To validate this prediction, we imaged WT and L119A mutant Parkin. We observed partial wild-type Parkin recruitment onto mitochondria upon MIRO1 overexpression (Fig. 5G,H), which we quantified as previously (Safiulina et al, 2019), as an increase in signal heterogeneity. Importantly, Parkin-L119A staining remained homogenously cytosolic even upon MIRO1 overexpression (Fig. 5G,H). Because the interaction between Parkin and Miro has been proposed to be important for Miro degradation (Wang et al, 2011) we assessed Miro's fate with Miro-binding-deficient Parkin-L119A. We find that FCCP-induced Miro degradation kinetics were comparable in WT and L119A mutant Parkin (Fig. EV5C). Moreover, Parkin activation, as observed by Parkin autoubiquitination and clustering onto damaged mitochondria was also comparable (Fig. EV5C,D). We conclude that the documented Miro-Parkin interaction is elicited through Parkin-L119 and the Miro-ELF pocket, but is not required for Miro degradation. The functional consequence of this interaction, therefore, remains to be understood.

## Discussion

### The various biochemical features affecting Miro binding

Here, we identify that Miro proteins interact with a variety of partners with a similar configuration, whereby interactors bind a hydrophobic pocket. Note that all predictions carried out with MIRO1 were performed with MIRO2, and no differences were observed (Dataset EV1).

Previous work has suggested that client binding is dependent on Miro's calcium and nucleotide status. The ELF pocket is made in part by the $Ca^{2+}$-binding EF hand. Interestingly, Drosophila Miro's crystal structure shows a density corresponding to an unknown small-molecule ligand occupying the ELF pocket, enlarging it, thus preventing $Ca^{2+}$ binding (Klosowiak et al, 2013). Whether this corresponds to a physiological ligand competing partners out of Miro-ELF pocket or is an artifact of protein expression is yet unknown, but these findings suggest that $Ca^{2+}$, ligand and client binding in the ELF pocket are mutually exclusive.

In addition to the ELF pocket, all clients except MYO19 and Parkin establish an antiparallel β-strand with Miro-GTPase1. The significance of this feature is difficult to assess experimentally as β-sheets do not obviously involve mutable side chains. Nonetheless, the coevolution of VPS13D's β-strand and the ELF-binding motif constitute strong evidence for the importance of this strand in Miro binding. Moreover, the β-sheet is established at a highly conserved patch of GTPase1 previously named the SELFYY surface (after a conserved peptide) (Smith et al, 2020). How nucleotide binding in the GTPase1 domain affects client binding is unclear, as the SELFYY interface is opposite to the nucleotide-binding pocket, and is largely unaffected by the nucleotide status of the protein (Smith et al, 2020). It is possible that nucleotide binding elicits larger allosteric changes; for instance, controlling the flexible hinge positioning the GTPase1 domain, which could control access to the ELF pocket.

Although the general Miro-binding configuration is shared, details are intriguingly different (e.g., leucine vs phenylalanine, with or without salt bridge, or β-sheet). For instance, a glycine residue preceding the leucine/phenylalanine is found in several clients. In

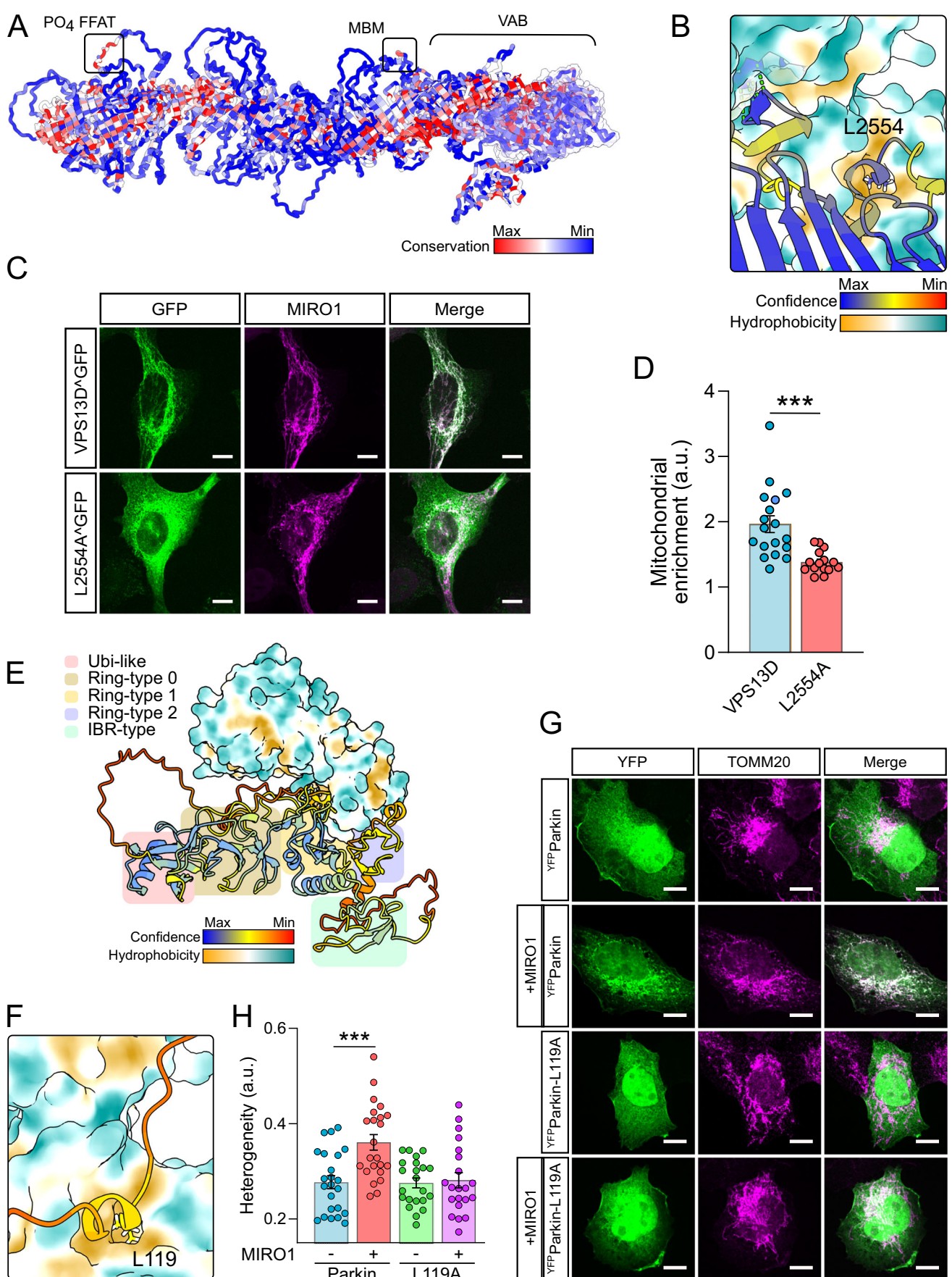

◀
**Figure 5.    Conserved leucine residues in both VPS13D and Parkin interact with the Miro-ELF pocket.**

(A) Predicted full-length structure of human VPS13D with residues colored by conservation. $PO_4$ FFAT phospho-FFAT motif for VAP binding, MBM Miro-binding motif, VAB  VPS13 adaptor binding domain. (B) AlphaFold2 multimer prediction of MIRO1 (surface) and VPS13D. (C) Representative images of internally GFP-tagged wild-type and L2554A mutant VPS13D (green) in U2OS cells overexpressing MIRO1 (magenta). (D) Quantification of mean mitochondrial intensity divided by mean intensity in the cytoplasm. $N = 15$–18 cells over three independent experiments. Statistical significance was calculated by an unpaired Student's *t* test. (E) Structural prediction of interaction between MIRO1 (surface) with full-length Parkin. Colored boxes highlight the individual predicted domains of Parkin. (F) Zoom of structural predictions of Miro-ELF pocket and Miro-binding motif of Parkin. (G) Representative images of wild-type and L119A [YFP]Parkin (green) in U2OS cells either with or without [myc]MIRO1 overexpression. Mitochondria were stained with TOMM20 antibody (magenta). (H) Quantification of the heterogeneity of YFP signal from wild-type and L119A Parkin, both with and without MIRO1 overexpression. Statistical significance was calculated by one-way ANOVA with Tukey post hoc test. Data information: All data are shown as mean ± SEM. *** is $P < 0.001$. Scale bars represent 10 µm. Source data are available online for this figure.

CENPF, this glycine is vital for binding (Peterka and Kornmann, 2019). Why it is not important in all clients might boil down to the slightly different conformations taken by the client's backbone when entering and exiting the ELF pocket. Chains with glycines harbor bond angles that other amino acids cannot adopt (Appendix Fig. S1). All of the Miro-binding motifs identified here are highly divergent to the point that it is challenging to predict Miro binding from the primary sequence alone. A survey of the mitochondrial proteome against a crude motif (FADI), identified new Miro-ELF pocket clients (namely, MTFR2, and subsequently MTFR1/1L); this search was neither sensitive (none of the known clients met these criteria) nor specific (of five hits, only MTFR2 was predicted by AlphaFold to bind Miro). The plasticity in binding motifs means more sophisticated searches are required to reach sensitivity and specificity. One explanation for flexibility in the motif is that it is an easily "evolvable" element that can likely be exploited when an interaction with Miro becomes a competitive advantage as exemplified by the appearance/disappearance of Miro-binding motifs in VPS13D through metazoan evolution (Fig. EV5A). A parallel might be drawn to the VAP proteins that bind short and diverse FFAT motifs to recruit proteins and whole organelles to the ER (Murphy and Levine, 2016). As such Miro might be regarded as a general and regulatable adaptor to recruit proteins and organelles to mitochondria.

## Miro proteins as coordinators of mitochondrial homeostasis

The identification of key leucines/phenylalanines in clients provides an opportunity to decipher the importance of their binding to Miro. Single leucine/phenylalanine point mutants provide a means to perturb the residue specifically required for Miro binding whilst keeping the rest intact, i.e., maintaining Miro-independent processes that would be lost with gene deletions. Indeed, we have shown that mutating CENPF-F2989 prevented recruitment to mitochondria but yielded surprisingly healthy mice (Peterka and Kornmann, 2019). It will be crucial to assess the phenotypic consequences of specifically disrupting interaction with Miro for other partners as well. For instance, Trak proteins are recruited to mitochondria independently of Miro (López-Doménech et al, 2018). What, therefore, is the specific role of Miro binding in their microtubule-dependent mitochondrial transport function?

A noteworthy consequence of a shared binding site on Miro for its clients, is that the roles of Miro in microtubule-dependent trafficking, actin dynamics, mitochondrial morphology, lipid transport and mitophagy must be competitive, further suggesting that there is, to some extent, competition between these processes themselves (Oeding et al, 2018). Previous groups have proposed elements of this idea, as in the model where mitochondria must be released from microtubules to be efficiently degraded (Hsieh et al, 2016; Ashrafi et al, 2014; Wang et al, 2011) or attached to actin filaments during early-, and microtubule tips during late-mitosis (Chung et al, 2016; Kanfer and Kornmann, 2016; Majstrowicz et al, 2021). This competition might now be traced at the molecular level to competitive binding. This mechanism is likely shared in many eukaryotic species, for a currently unknown number of processes at mitochondria. For instance, we do not know any client for plant Miro. We therefore expect that Miro function is to be a central point at the outer mitochondrial membrane to coordinate mitochondrial homeostasis.

# Methods

## DNA constructs

List of all plasmids used in this study can be found in Table 1.

## Antibodies and dyes

Primary antibodies: mouse anti-myc (9E10 at 1:1000), mouse anti-Flag (M2 at 1:1000), rabbit-TOMM20 (Santa Cruz—sc-11415, 1:500), rabbit anti-myc (Abcam—ab9106, 1:1000), mouse anti-GFP (Merck— 11814460001 1:2000), mouse anti-actin (Merck—A2228, 1:2000). Secondary antibodies: donkey anti-mouse IgG H&L-AlexaFluor-647 (Abcam ab1501017 at 1:500), donkey anti-rabbit IgG H&L-AlexaFluor-568 (Abcam ab175470 at 1:500). MitoTracker Orange CMTMRos was obtained from Thermo Fisher Scientific (M7510).

## Yeast and mammalian cell lines

### Generating yeast strains
MDM34-mCherry::HIS3 and internally GFP-tagged Gem1 (at position 263) were generated previously (Kornmann et al, 2009; English et al, 2020). A L248A mutation in MDM34 was generated using the CRISPR/Cas9 system (Hu et al, 2018) using the following gRNA: 5'-tttcaagcattgtgtcgtcgagg-3' and a repair template including the desired mutation. All yeast strains used can be found in Table 2.

### Mammalian cells
U2OS, HeLa, and Cos7 cells were cultured in DMEM with 4.5 g/L glucose plus 10% fetal bovine serum, GlutaMAX, and penicillin/

**Table 1.  DNA constructs used in this study.**

| | |
|---|---|
| pEG202-CENPF42 | Kanfer et al, 2015 |
| pEG202-CENPF42-D2991R | This study |
| pJG4-5-MIRO1-1-594 | Kanfer et al, 2015 |
| pJG4-5-MIRO1-1-400 | This study |
| pJG4-5-MIRO1-181-594 | This study |
| pJG4-5-MIRO1-1-280 | This study |
| pJG4-5-MIRO1-181-400 | This study |
| pJG4-5-MIRO1-281-594 | This study |
| pJG4-5-MIRO1-1-180 | This study |
| pJG4-5-MIRO1-181-280 | This study |
| pJG4-5-MIRO1-281-400 | This study |
| pJG4-5-MIRO1-401-594 | This study |
| pJG4-5-MIRO1-R261D | This study |
| pJG4-5-MIRO1-R263D | This study |
| pJG4-5-MIRO1-R261D/R263D | This study |
| pEG202-Trak1-577-620 | This study |
| pEG202-Trak1-577-620-L594 | This study |
| pEG202-Trak1-577-620-D599R | This study |
| pEG202-MYO19-919-970 | This study |
| pEG202-MYO19-919-970-F948A | This study |
| pEG202-MYO19-919-970-E954R | This study |
| pEGFP-C1-MYO19 | (Oeding et al, 2018) Addgene: #134987 |
| pEGFP-C1-MYO19-F948A | This study |
| pEGFP-C1-Trak1 | (Birsa et al, 2014) Addgene: #127621 |
| pEGFP-C1-Trak1-L594A | This study |
| pRK5-myc-MIRO1 | (Fransson et al, 2003) Addgene: #47888 |
| pFRT/TO-3flag6his-MIRO1 | Kanfer et al, 2015 |
| pEGFP-C1-MIRO1 | Birsa et al, 2014 |
| pCMV6-Mtfr1-myc-DDK | Origene MR204817 |
| pCMV6-Mtfr1-F76A-myc-DDK | This study |
| pCMV6-Mtfr2-myc-DDK | Origene MR205532 |
| pCMV6-Mtfr2-F93A-myc-DDK | This study |
| pCMV6-Mtfr1l-myc-DDK | Origene MR203935 |
| pEGFP-N1-MTFR1L | This study |
| pEGFP-N1-MTFR1L-L62A | This study |
| pEG202-Mtfr1 | This study |
| pEG202-Mtfr1-F76A | This study |
| pEG202-Mtfr2 | This study |
| pEG202-Mtfr2-F93A | This study |
| pEG202-MTFR1L | This study |
| pEG202-MTFR1L-L62A | This study |
| pCMV-VPS13D^GFP | (Guillén-Samander et al, 2021) Addgene: #174109 |
| pCMV-VPS13D-L2554A^GFP | This study |
| pEYFP-C1-Parkin | (Narendra et al, 2008) Addgene: #23955 |
| pEYFP-C1-Parkin-L119A | This study |

**Table 2.  Yeast strains used in this study.**

| | |
|---|---|
| EGY48 | Yeast two-hybrid competent yeast. |
| ByK302 | By4741 *MDM34*-mCherry::His3; Gem1_263GFP |
| ByK2029 | ByK302 with L248A mutation in *MDM34* by CRISPR/Cas9 mutagenesis |

streptomycin. Wild-type and Miro1/2 double knockout mouse embryonic fibroblasts (MEFs), characterized previously (López-Doménech et al, 2018), were cultured in DMEM with 4.5 g/L glucose plus 15% fetal bovine serum, GlutaMAX, and penicillin/streptomycin. For fixed imaging, MEFs were seeded on fibronectin-coated coverslips.

## Yeast two hybrid

All yeast two-hybrid assays were based on LexA fusion proteins. EGY48 yeast were transformed with a pJG4-5 MIRO1 construct (prey) and a bait-containing plasmid (pEG202). For growth assays, yeasts were streaked on SC-Leu+Gal media and grown at 30 °C. For fluorescence yeast two-hybrid assays a modified protocol from (Plovins et al, 1994) was used. Yeasts were grown overnight in SC-Trp-His-Ura + 2% raffinose and 0.2% glucose and then switch to overnight in SC-Trp-His-Ura + 2% galactose. The following day, 2,000,000 cells for each condition were collected and resuspended in ice-cold 70% ethanol and shaken at 2,850 rpm for 5 min to permeabilize the cells. Cells were then pelleted and resuspended in 10 ml buffer Z (0.06 M $Na_2HPO_4$, 0.04 M $NaH_2PO_4$, 0.01 M KCl, 0.001 M $MgSO_4$ and 0.27% 2-mercaptoethanol) for CENPF and Trak1 and 1 ml of buffer Z for MYO19 due to differences in signal intensity. 50 μl of cell suspension and 50 μl of Fluorescein di-beta-D-galactopyranoside (FDG; 0.5 mg/ml dissolved in 98% water, 1% ethanol and 1% DMSO; Stratatech—14001) were then mixed together and imaged using the Fluorescein-FITC channel on an iBright-FL1500. Data are well fluorescence minus signal for empty vector divided by mean signal over all wells.

## Structural predictions

All structure figures were generated in ChimeraX (Pettersen et al, 2021).

### AlphaFold predictions
Monomeric Mdm34 and MIRO1 AlphaFold2 predictions were obtained from the AlphaFold-European Bioinformatics Institute database. Protein–protein interaction predictions were made using the AlphaFold2 multimer model (Evans et al, 2021)—ran both remotely and on the open-source AlphaFold.ipynb on Google Colab.

### VPS13D
A full-length VPS13D structure was predicted using a coarse-grained molecular dynamics simulation (MoDyFing) to fold the 3D structure of VPS13D from its primary sequence depending on the structural constraints (residue distances and torsion angles) that are inferred by deep-learning methods. The torsion angles (phi and psi) were predicted by the ESIDEN tool (Xu et al, 2021), while the distance between pairwise residues was inferred by the ProSpr mode (Stern et al, 2021). The constraints were used to predict protein 3D structure of no more than 900 residues. As such, VPS13D was split into seven fragments including three overlapping fragments. We leveraged the MoDyFing tool to fold each fragment using the inferred constraints and implemented the MODELLER tool to assemble the predicted structures of the seven fragments.

*Mapping conservation of amino acids*
Residue conservation for MIRO1 and VPS13D was made using the top 1000 conserved sequences in comparison to the human protein.

## Bioinformatics

Multiple sequence alignments were generated with MUSCLE and displayed with Jalview. Ramachandran plots used density data from (Lovell et al, 2003).

## Fluorescence microscopy

*Live imaging of yeast*
Yeast-saturated cultures were reseeded to OD of 0.1 in YPD and left to recover for 6 h. Roughly 500,000 cells were then washed in SC media and plated on a microscope slide with a coverslip on top. Images were obtained using a IX81 Olympus inverted spinning disk microscope with an EM-CCD camera (Hamamatsu Photonics) using a 100× oil objective (NA = 1.4).

*Fixed imaging of mammalian cells*
Cells were fixed with 4% paraformaldehyde in PBS for 10 min at room temperature and blocked with 5 mg/ml bovine serum albumin, 10% horse serum and 0.2% Triton X100 diluted in PBS. Primary and secondary antibodies were diluted in blocking buffer and used to stain cells for 1 h at room temperature. Images were taken on either a IX81 Olympus inverted spinning disk microscope with an EM-CCD camera (Hamamatsu Photonics) using a 100× oil objective (NA = 1.4) or a Zeiss LSM700 confocal using a 63× oil objective (NA = 1.4).

## Image analysis

No statistical method was used to determine sample size. Generally, samples were not blinded, but all quantifications were performed automatically (usually with bespoke ImageJ scripts).

*Mitochondrial enrichment*
Mitochondrial enrichment of fluorescent signal was calculated by dividing the mean fluorescence overlapping with a thresholded mitochondrial marker (e.g., Tom20 or mtDsRed) divided by the mean fluorescence intensity in the rest of the cell. For VPS13D, due to the high intensity of VPS13D^GFP signal at the Golgi, blind analysis was performed on 8 μm$^2$ crops of cells at a point where mitochondria are tubular and away from the perinuclear GFP signal.

*ERMES enrichment*
Gem1 enrichment at ERMES was calculated as the integrated density of signal overlapping with Mdm34-mCherry divided by the integrated density of GFP signal in the whole cell. The cell was identified using the YeastMate plugin (Bunk et al, 2022) in ImageJ.

*Heterogeneity of Parkin signal*
Data were blinded and quantified by taking 8 μm$^2$ crops of cells at a point where mitochondria are tubular and away from the nucleus, using the TOMM20 stain for reference. The coefficient of variation of YFP-Parkin signal was then calculated by dividing the standard deviation of YFP signal intensity by the mean intensity.

## Data availability

All data are available in the main text or the Supplementary Materials.

## Peer review information

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

## Acknowledgements

The authors are grateful for manuscript feedback from Agnès Michel and Viktoriya S. Toncheva. The authors also gratefully acknowledge the Micron Advanced Bioimaging Facility (supported by Wellcome Strategic Awards 091911/B/10/Z and 107457/Z/15/Z) for their support & assistance in this work. This work was funded by Wellcome Trust grant 214291/Z/18/Z (awarded to BK), Wellcome Trust grants 223202/Z/21/Z and 222/519/Z/21/Z (awarded to JTK) and an MRC PhD studentship 2397875 (CODT).

## Author contributions

**Christian Covill-Cooke**: Conceptualization; Supervision; Investigation; Visualization; Methodology; Writing—original draft; Writing—review and editing. **Brian Kwizera**: Conceptualization; Investigation; Methodology; Writing—review and editing. **Guillermo López-Doménech**: Conceptualization; Investigation; Methodology; Writing—review and editing. **Caleb OD Thompson**: Conceptualization; Investigation; Methodology; Writing—review and editing. **Ngaam J Cheung**: Conceptualization; Supervision; Investigation; Methodology. **Ema Cerezo**: Conceptualization; Investigation; Methodology. **Martin Peterka**: Conceptualization; Investigation; Methodology; Writing—review and editing. **Josef T Kittler**: Conceptualization; Supervision; Funding acquisition; Methodology; Project administration; Writing—review and editing. **Benoit Kornmann**: Conceptualization; Supervision; Funding acquisition; Investigation; Visualization; Methodology; Writing—original draft; Project administration; Writing—review and editing.

## Disclosure and competing interests statement

The authors declare no competing interests.

# Expanded View Figures

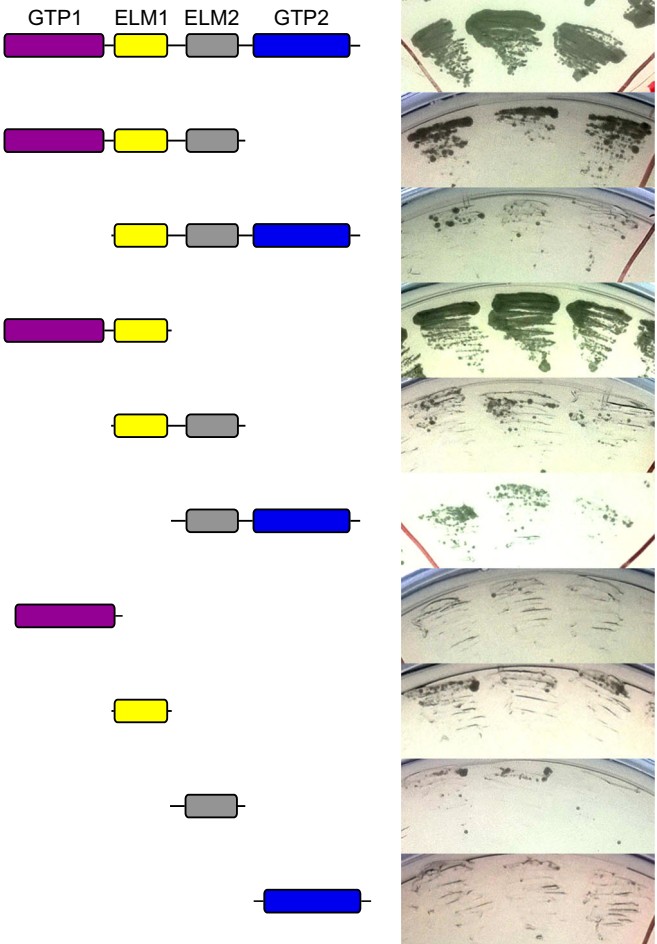

**Figure EV1. GTPase1 and ELM1 of MIRO1 are necessary and sufficient for CENPF binding.**

Representative yeast two-hybrid of CENPF-2819-3114 (bait) with MIRO1 truncations (prey). Each streak is from an independently generated strain. Growth of the second fragment was less robust than that of the first and fourth fragments because the expression of this fragment was poor.

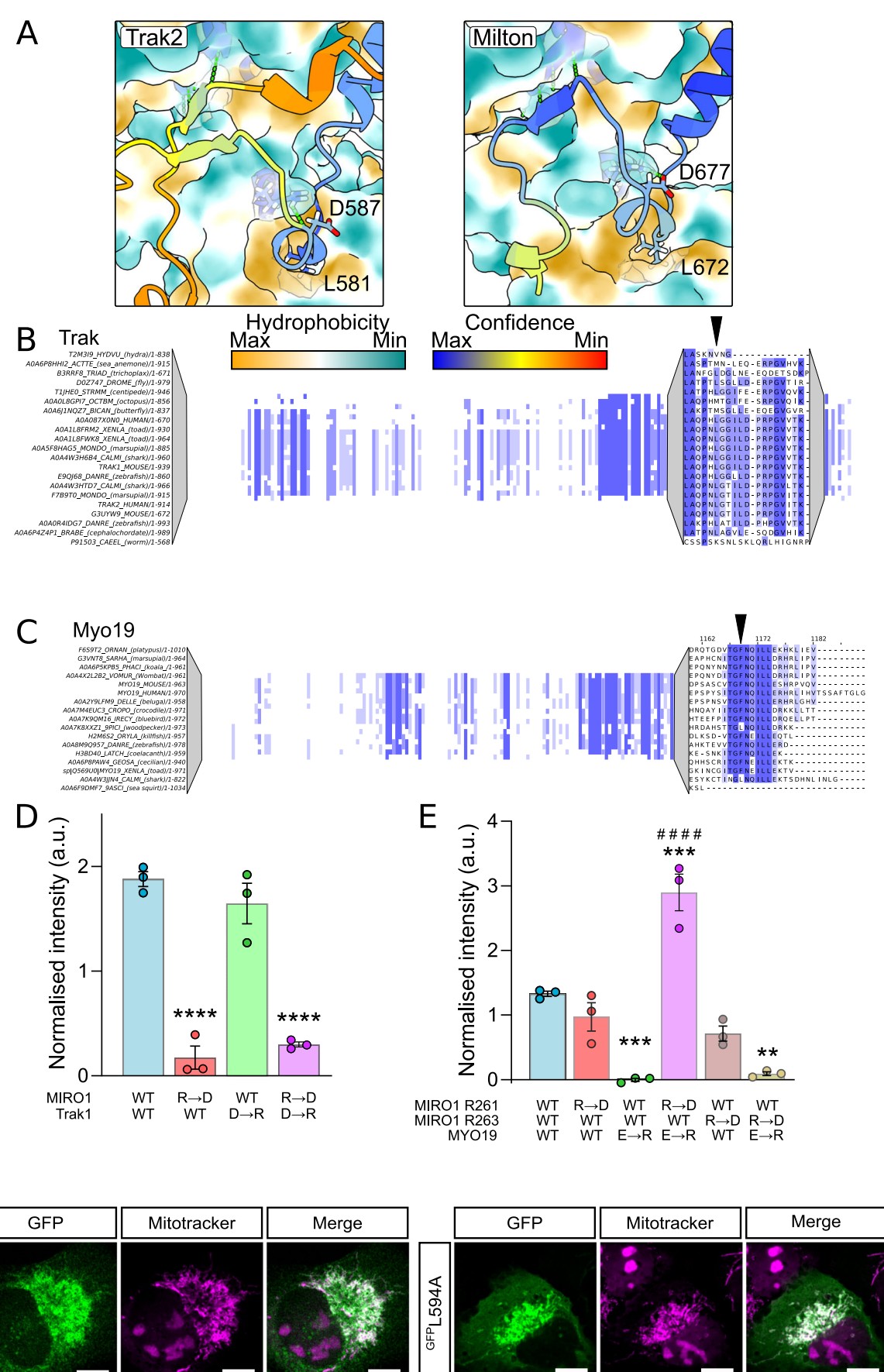

**Figure EV2.  Conserved features in Trak1 and MYO19 that interact with MIRO1.**

(A) AlphaFold2 multimer predictions of human Trak2 with MIRO1 (surface) or *Drosophila melanogaster* Miro (surface) and Milton (Trak orthologue). Color coding as in Fig. 1D. (B) Sequence alignment of Trak orthologues around the Miro-binding motif. (C) Sequence alignment of MYO19 orthologues around the Miro-binding motif. (D) Quantification of fluorescence yeast two-hybrid of wild-type and charge swapped mutants of MIRO1 (prey) and mouse Trak1-577-620 (bait). R → D and D → R are MIRO1-R263D and Trak1-D599R, respectively, $n = 3$. (E) Quantification of fluorescence yeast two-hybrid of wild-type and charge swapped mutants of MIRO1 (prey) and MYO19-919-970 (bait). R → D and E → R are MIRO1-R261D/MIRO1-R263D and MYO19-E954R, respectively, $n = 3$. (F) Representative images of wild-type and L594A mouse $^{GFP}$Trak1 (green) in U2OS cells. Mitochondria are stained with Mitotracker-Orange (magenta). Scale bars represents 10 μm. Data information: D and E statistical significance was calculated by one-way ANOVA with Tukey post hoc test. **, *** and **** denotes $P < 0.01$, $0.001$ and $0.0001$ in comparison to WT conditions. #### is $P < 0.0001$ in comparison to WT-MIRO1 + MYO19-E → R.

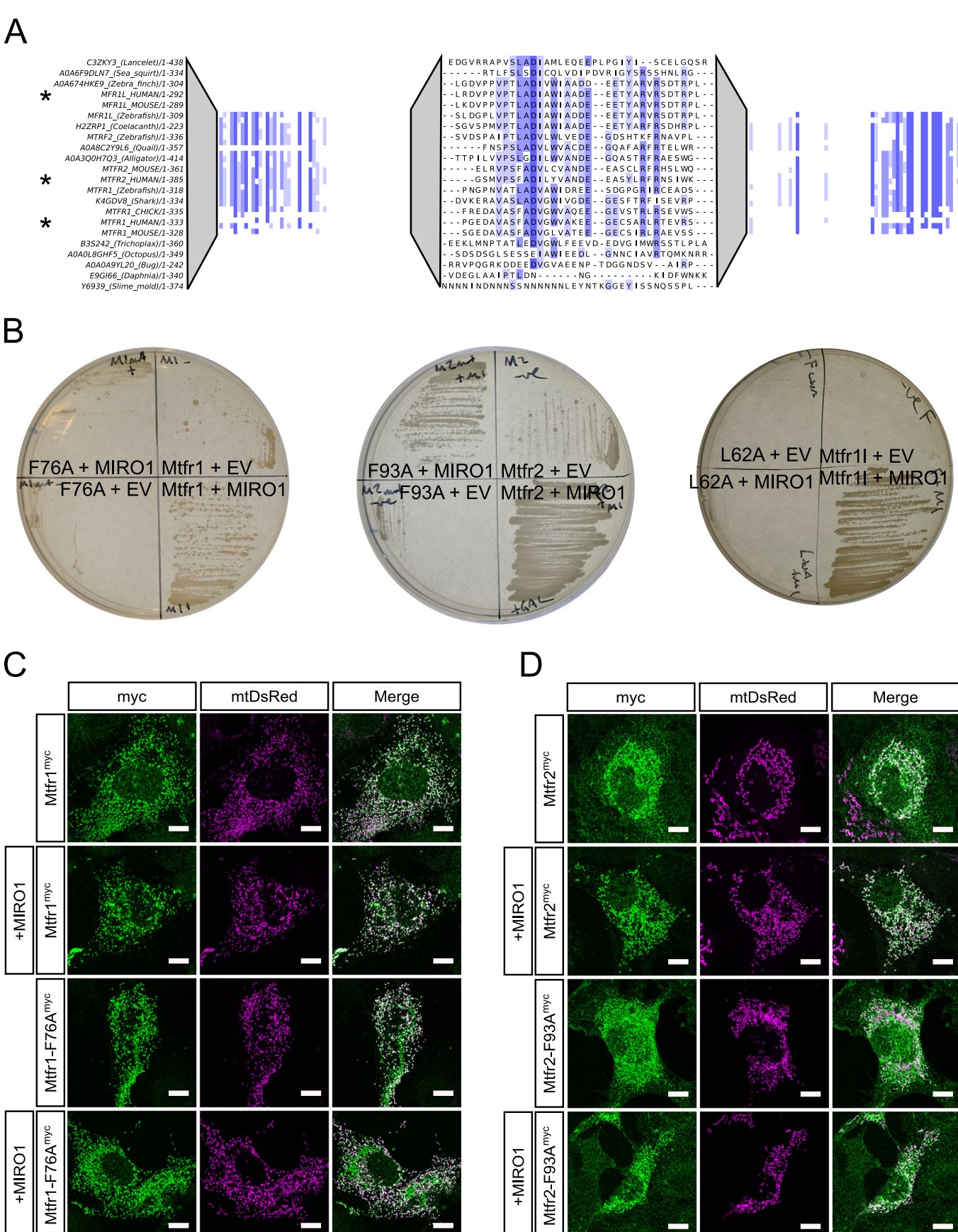

◀ **Figure EV3. Mtfr1/2/1L interact with MIRO1 via conserved motif.**

(A) Sequence alignment of sequences of MTFR1, MTFR2 and MTFR1L surrounding the Miro-binding motif. (B) Yeast two-hybrid growth assay of MIRO1 (prey) with wild-type or point mutants of full-length Mtfr1, Mtfr2 and Mtfr1l. EV means empty vector control. (C) Representative images of U2OS Cos7 cells expressing either wild-type or F76A point mutated Mtfr1 (green) both with or without ᴳᶠᴾMIRO1 overexpression. Mitochondria are stained with mtDsRed (magenta). (D) Representative images of U2OS Cos7 cells expressing either wild-type or F93A point mutated Mtfr2 (green) both with and without ᴳᶠᴾMIRO1 overexpression. Mitochondria are stained with mtDsRed (magenta). Scale bars depict 10 μm.

# A

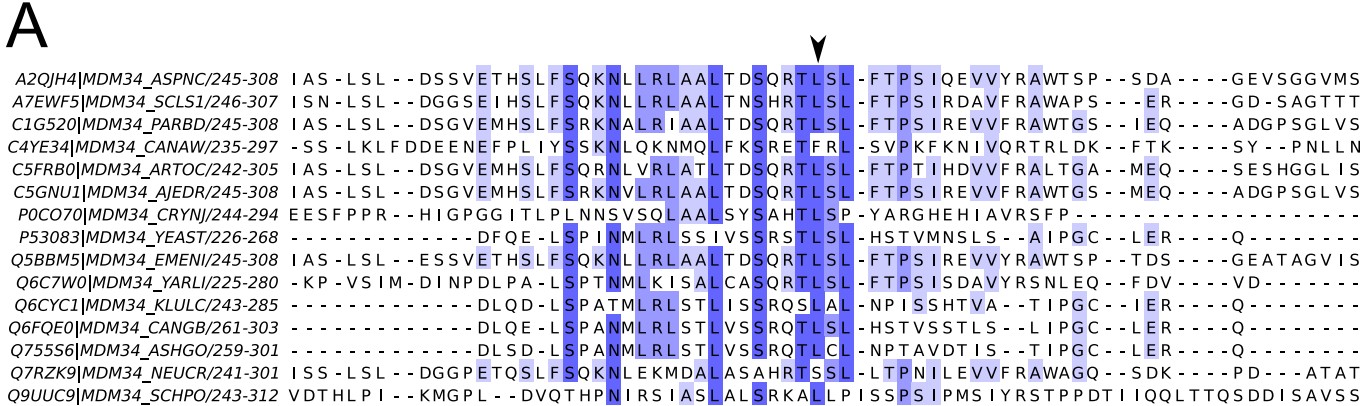

# B

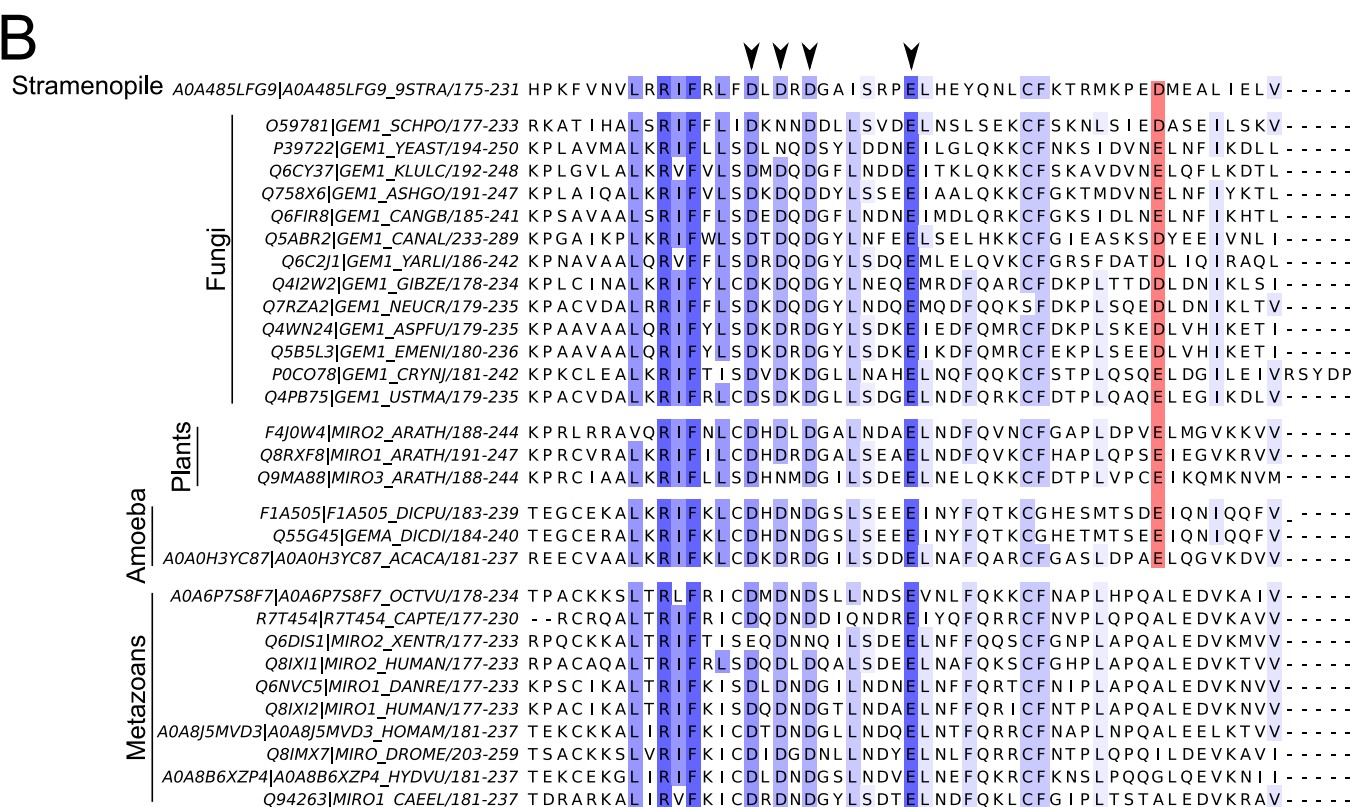

**Figure EV4.  Conservation of Gem1 and Mdm34.**

(**A**) Sequence alignment of Mdm34. Arrowheads indicate the position of the Leucine inserted in the ELF. (**B**) Sequence alignments of Miro orthologues in fungi, plants, amoeba, and metazoans. Arrowheads indicate the acidic residues coordinating Ca$^{2+}$ in the EF hand. Red highlights key acidic residue not found in metazoans that is likely required for Mdm34 binding.

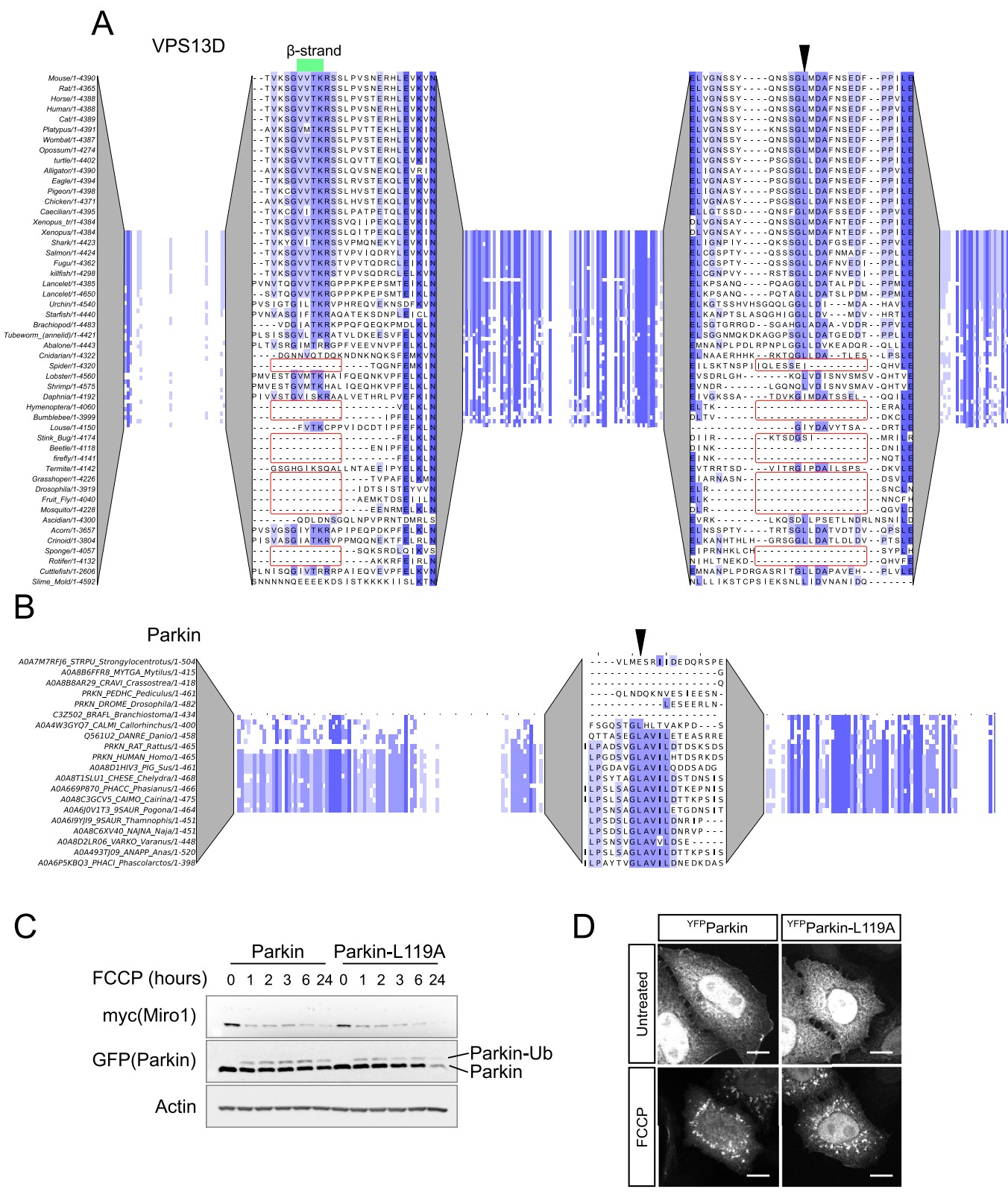

◄ **Figure EV5. Conservation of VPS13D and Parkin Miro-binding motifs.**

(A) Sequence alignment of VPS13D. Red boxes show sequences that lack both the β-strand and the Leucine-containing motif of the MBM together. (B) Sequence alignments of Parkin. Arrowheads point to the conserved leucine residues mutated in this study. (C) HeLa cells transfected with WT or L119A mutant Parkin and treated with 10 μM FCCP to induce mitochondrial depolarization. Timepoints collected as indicated were analyzed by western blotting using the indicated antibodies. (D) YFP-Parkin signal in untreated and 10 μM FCCP treated (1 h) HeLa cells. Scale bar is 10 μm.

