## [Peer Review File · The EMBO Journal]

Shared Structural Features of Miro Binding Control Mitochondrial Homeostasis

Benoit Kornmann, Christian Covill-Cooke, Brian Kwizera, Guillermo López-Doménech, Caleb Thompson, Ngaam Cheung, Ema Cerezo, Martin Peterka, and Josef Kittler

DOI: [10.15252/emboj.2023115419](https://doi.org/10.15252/emboj.2023115419)

Corresponding author(s): Benoit Kornmann (benoit.kornmann@bioch.ox.ac.uk)

Review Timeline:

Submission Date:	25th Aug 23
Editorial Decision:	3rd Sep 23
Appeal Received:	5th Sep 23
Editorial Decision:	19th Oct 23
Revision Received:	13th Nov 23
Editorial Decision:	12th Dec 23
Revision Received:	14th Dec 23
Accepted:	15th Dec 23

Editor: William Teale

Transaction Report:

Dear Benoît,

Thank you for submitting your manuscript entitled 'Shared Structural Features of Miro Binding Control Mitochondrial Homeostasis' (EMBOJ-2023-115419) to our editorial office. We have now considered the study within our editorial team, and unfortunately come to the conclusion that we cannot offer publication in The EMBO Journal. You present a detailed analysis of the molecular relationship between Miro and Cenp-F: an interaction which you have previously shown to direct the distribution of mitochondria during mitosis. Your work expands this analysis to screen for, identify and characterize other binding conformations, finding a pocket which mediates the interaction of Miro1 with diverse interactors. In this context, your comparison of Gem1 and Miro1 was particularly intriguing. However, we are not convinced that, without a more quantitative analysis of the interactions you describe (particularly in terms of key cooperative or competitive interactions), your analysis of this relationship provides sufficient mechanistic depth to make the study a strong candidate for publication in a broad general journal like The EMBO Journal at this stage.

However, given the potential value of your results to researchers directly in the field, we feel that the study should be well-suited for Life Science Alliance (<http://www.life-science-alliance.org/>), our open access journal launched in partnership with Rockefeller University Press and Cold Spring Harbor Laboratory Press. LSA aims to publish solid findings of value to particular communities across all areas of biology. I have therefore briefly discussed the work with the editors of Life Science Alliance, who would indeed be pleased to send your work for in-depth external review, without need for prior reformatting. Should you be interested in this option, please simply follow the link below for transfer. Eric Sawey, Executive Editor of Life Science Alliance (e.sawey@life-science-alliance.org), will be happy to answer any questions you may have.

I am sorry that I cannot be more positive for The EMBO Journal on this occasion, but very much hope that you will find this transfer option worthwhile.

Best wishes,

William

William Teale, PhD
Editor
The EMBO Journal
w.teale@embojournal.org

** As a service to authors, EMBO Press provides authors with the possibility to transfer a manuscript that one journal cannot offer to publish to another EMBO publication or the open access journal Life Science Alliance launched in partnership between EMBO Press, Rockefeller University Press and Cold Spring Harbor Laboratory Press. The full manuscript and if applicable, reviewers' reports, are automatically sent to the receiving journal to allow for fast handling and a prompt decision on your manuscript. For more details of this service, and to transfer your manuscript please click on Link Not Available. **

Dear William,

thanks for taking the time to read our work. I am getting back to you because we are very puzzled by your decision. I wonder if we could have 10 min to chat over it on the phone.

In a nutshell, we think that our study offers a compelling set of answers in a field that is at the moment quite messy. We like to keep our study at the purely mechanistic and structural level. Although we are aware of the physiological implications of our finding in terms of competitive binding (which we discuss), we do not think that we should engage in these waters at this point as, again, we are trying to clarify the field with a compelling finding, not add more mess, which is inevitably what will happen if we engage with the more physiological implications of our findings. Indeed, there are thousands of cell types, each expressing a set amount of Miro and of each client. Anything we could possibly find there in term of competitive binding will never be as unattackable and generalizable (as you say, true in yeast, true in humans) as what we have already shown. We think that our story as it stands now is very likely to cruise through peer-review, in large part because it does limit itself to what can be truly ascertained, without trying to make grand (and precariously supported) claims.

I am available today and most of the rest of the week and would love to chat about it, to help us understand what more you would want to see in such a study.

Looking forward to hearing from you and best wishes

Benoit

Dear Benoît,

Thank you again for the submission of your manuscript entitled "Shared Structural Features of Miro Binding Control Mitochondrial Homeostasis", (EMBOJ-2023-115419) We have now received reports from three referees, which I copy below.

As you can see from their comments, while the report of referee #3 mirrored my initial thought that a level of quantification of the ELF binding pocket-client interactions would significantly enhance the manuscript, referees #1 and #2 expressed the opinion that your work is sufficiently compelling without this extra level of detail.

Therefore, I would like to invite you to address the comments of all referees in a revised version of the manuscript. I should add that it is The EMBO Journal policy to allow only a single major round of revision and that it is therefore important to resolve the main concerns at this stage. I believe the concerns of the referees are reasonable and addressable, but please contact me if you have any questions, need further input on the referee comments or if you anticipate any problems in addressing any of their points. Please, follow the instructions below when preparing your manuscript for resubmission.

I would also like to point out that as a matter of policy, competing manuscripts published during this period will not be taken into consideration in our assessment of the novelty presented by your study ("scooping" protection). We have extended this 'scooping protection policy' beyond the usual 3 month revision timeline to cover the period required for a full revision to address the essential experimental issues. Please contact me if you see a paper with related content published elsewhere to discuss the appropriate course of action.

Again, please contact me at any time during revision if you need any help or have further questions.

Thank you very much again for the opportunity to consider your work for publication. I look forward to your revision.

Best regards,

William

William Teale, Ph.D.
Editor
The EMBO Journal

When submitting your revised manuscript, please carefully review the instructions below and include the following items:

2) individual production quality figure files as .eps, .tif, .jpg (one file per figure).

3) a .docx formatted letter INCLUDING the reviewers' reports and your detailed point-by-point response to their comments. As part of the EMBO Press transparent editorial process, the point-by-point response is part of the Review Process File (RPF), which will be published alongside your paper.

4) a complete author checklist, which you can download from our author guidelines ([https://wol-prod-cdn.literatumonline.com/pb-assets/embo-site/Author Checklist%20-%20EMBO%20J-1561436015657.xlsx](https://wol-prod-cdn.literatumonline.com/pb-assets/embo-site/Author%20Checklist%20-%20EMBO%20J-1561436015657.xlsx)). Please insert information in the checklist that is also reflected in the manuscript. The completed author checklist will also be part of the RPF.

6) We require a 'Data Availability' section after the Materials and Methods. Before submitting your revision, primary datasets produced in this study need to be deposited in an appropriate public database, and the accession numbers and database listed under 'Data Availability'. Please remember to provide a reviewer password if the datasets are not yet public (see <https://www.embopress.org/page/journal/14602075/authorguide#datadeposition>). If no data deposition in external databases is needed for this paper, please then state in this section: This study includes no data deposited in external repositories. Note that

the Data Availability Section is restricted to new primary data that are part of this study.

Note - All links should resolve to a page where the data can be accessed.

8) For data quantification: please specify the name of the statistical test used to generate error bars and P values, the number (n) of independent experiments (specify technical or biological replicates) underlying each data point and the test used to calculate p-values in each figure legend. The figure legends should contain a basic description of n, P and the test applied. Graphs must include a description of the bars and the error bars (s.d., s.e.m.).

9) We would also encourage you to include the source data for figure panels that show essential data. Numerical data can be provided as individual .xls or .csv files (including a tab describing the data). For 'blots' or microscopy, uncropped images should be submitted (using a zip archive or a single pdf per main figure if multiple images need to be supplied for one panel). Additional information on source data and instruction on how to label the files are available at .

10) We replaced Supplementary Information with Expanded View (EV) Figures and Tables that are collapsible/expandable online (see examples in <https://www.embopress.org/doi/10.15252/embj.201695874>). A maximum of 5 EV Figures can be typeset. EV Figures should be cited as 'Figure EV1, Figure EV2" etc. in the text and their respective legends should be included in the main text after the legends of regular figures.

12) Our journal encourages inclusion of *data citations in the reference list* to directly cite datasets that were re-used and obtained from public databases. Data citations in the article text are distinct from normal bibliographical citations and should directly link to the database records from which the data can be accessed. In the main text, data citations are formatted as follows: "Data ref: Smith et al, 2001" or "Data ref: NCBI Sequence Read Archive PRJNA342805, 2017". In the Reference list, data citations must be labeled with "[DATASET]". A data reference must provide the database name, accession number/identifiers and a resolvable link to the landing page from which the data can be accessed at the end of the reference. Further instructions are available at .

Further instructions for preparing your revised manuscript:

We realize that it is difficult to revise to a specific deadline. In the interest of protecting the conceptual advance provided by the work, we recommend a revision within 3 months (17th Jan 2024). Please discuss the revision progress ahead of this time with the editor if you require more time to complete the revisions. Use the link below to submit your revision:

Referee #1:

Miro/Gem1 is an unconventional GTPase of the outer mitochondrial membrane that was shown to function as an adaptor for several proteins implicated in mitochondrial dynamics and traffic. Mechanistic details of the interaction of Miro with its client proteins had remained unclear. This manuscript fills this gap. The authors identify a hydrophobic pocket in Miro's Ca²⁺ binding domain (the ELF pocket) that mediates these interactions and the consensus motif present in client proteins that bind Miro. They go one to reveal molecular features of these interactions with a synergistic use of yeast genetics/cell biology, AlphaFold predictions, and studies in mammalian cells. Their conclusions are validated by insightful and compelling mutagenesis experiments. A key conclusion is the competitive nature of these interactions, suggesting that Miro can switch its partners in response to regulatory mechanisms. This is a good manuscript appropriate for publication in EMBO.

I have only minor comments.

The yeast two hybrid assay shows that CENPF D2991R impairs recruitment by Miro (Fig. 1F). Does this mutant also impair recruitment by Miro when expressed in mammalian cells?

Are there functional consequences for the mutations identified in this work? For example, does the mutation of CENPF (Fig. 1F) impair the reported redistribution of mitochondria by Miro and CENP-F to the cell periphery (Kanfer et al., Nat Comm 2015) or does the mutation of Parkin (Fig. 5H) prevent degradation of Miro after mitochondrial damage (Wang et al., Cell 2011)?

The quality of Fig. 5C is not optimal and does not convey the message that it is supposed to convey: reduced recruitment to mitochondria of VPS13D^ΔGFP L2554A. This figure should be replaced with a better figure

Referee #2:

Work presented in by Covill-Cooke et al reports a protein interaction interface between the MIRO GTPases and a series of binding partners (termed "clients" in the manuscript). The binding partners include CENPF, Trak, MYO19, MTFR1/2/1L, VPS13D and Parkin, and some experiments were done with the yeast orthologue interaction with Mdm34. The primary analysis was initiated using AlphaFold2 to identify a potential binding pocket, and some of the interactions were tested using the yeast two hybrid method. The structural modeling of the interacting face led them to generate targeted point mutations to abolish the interactions (2 hybrid) and potential effects on the recruitment of the cytosolic interacting proteins to mitochondria were seen with fluorescent imaging. The binding pocket in Miro was identified as a short hydrophobic region in the first binding pocket, or the EF hand domain with flanking helices (ELM1 domain). That Miro binds so many proteins through the same domain (albeit with some differences in the specific residues of the domain) lead authors to propose Miro as a universal mitochondrial adaptor protein.

Strengths:

The AlphaFold predictions allowed them to map the binding interface and rationally design point mutations that may abolish the interactions. The authors previously mapped the C-terminal 42 amino acids of CENPF as the Miro binding domain, which was used as bait in the Y2H against a series of truncation mutants of Miro. This narrowed the region of Miro to the GTPase and ELM1 domain, and AlphaFold2 identified a predicted conserved patch where a phenylalanine inserts into a pocket within ELM1, opposite to the Calcium binding domain (a patch on Miro they call the ELF pocket), and a salt bridge was defined as important for binding. Similar analysis was done with domains previously identified as requisite for Miro binding in Trak2 and MYO19. These analysis underscore the power of AlphaFold2 in mapping protein interaction interfaces, which were validated here by Y2H approaches. With the specific residues mapped in 3 of Miro's binding partners, the authors searched MitoCarta3.0 for additional Miro ELF-pocket binding partners containing this conserved motif they call FADI. (It was not entirely clear why they would search MitoCarta as their only approach given the major known binding partners are cytosol.) From this they identify MTFR1/2/1L as candidates, and confirmed the interactions using Y2H (and in supplement), and also looked at VPS13 and Parkin. Moreover, authors looked for evolutionary confirmation of these interfaces in yeast orthologues of Miro, called GEM1, with similar results.

Weaknesses:

The motivation of this study was to understand how Miro GTPases may be able to bind so many reported "clients", whether this was a competitive type of event, or if larger complexes may form, etc. The weakness then comes in the lack of biochemical confirmation of these interactions, and how they may actually compete functionally. The recruitment of cytosolic proteins (CENPF, Trak2, MYO19) to mitochondria may be mildly affected in Miro overexpression studies, but this is not proof that the effects are indeed due to the diminished Miro interactions. There are no functional assays to determine the effects of these processes on mitochondrial motility, or mitophagy, or whichever additional functions they envision. The effects on VAPB recruitment seem rather minor (Fig 5D) and it's not clear this is proof that the consequence is due exclusively to an inhibition of Miro binding.

Perhaps most confusing is their assertion that this interacting face forms a meaningful, functional interaction with MTFR1/2/1L (also called MTP18). This is an established inner membrane protein with three transmembrane domains that was first studied in the mid-2000s. It is difficult to conceive how it could interact with a cytosol-facing domain on Miro. The Miro1/2 double KO cells see normal import of the MTFR1 and 2 proteins, therefore this interaction is not essential to ensure the biogenesis of these inner membrane proteins. On the other hand, loss Miro1/2 did affect MTFR1L import, so perhaps this variant is not in the inner membrane? But this was not tested or followed in any functional way. The overexpression shows the mutant MTFRs accumulate in cytosol, however this may reflect structural alterations that abrogate protein targeting and insertion, rather than reflect functional binding to Miro. The targeting sequence of multispinning inner membrane proteins is not always so obvious and could have been interrupted by these mutations. The authors didn't mention the native topology or mitochondrial location of MTFP proteins, which was confusing. Is the MTFR1L76A (and F residue equivalents in MTFR2 and 1L) upstream of the first transmembrane domain (?) in a rather disordered region? I'm really not clear what the authors think this interaction with Miro is for, does it have to do with mitochondrial motility, or import somehow? it just seems very out of context and preliminary.

Summary:

I have not structured my review into specific experiments for the authors to address, rather I leave it to them to consider my comments. I am a strong believer in the transformative power of AlphaFold analysis in the identification of protein interaction interfaces. The authors have clearly made significant headway in understanding how Miro can bind multiple partners. The weakness is in the validation of their data, and lack of biochemical (with endogenous) approaches, or functional impacts. It is a broad assumption to assume the data coming from a Y2H interaction can be used to infer causality in the context of complex events on the mitochondrial surface where Miro binds many proteins, calcium and GTP/GDP. I do not think the authors must confirm function in all cases here, or to dig deeper into the role of calcium and GTPase activities in modulating these interactions. I understand that these are large questions. But more must be done to convince me that the essential residues mapped in the Y2H interactions are truly meaningful in the action of these Miro clients at mitochondria.

Referee #3:

This manuscript reports the conserved interaction between the mitochondrial protein Miro1 and its cytosolic clients. It proposes that the interfaces between Miro and its interaction partners include of a hydrophobic binding pocket in Miro, termed the "ELF pocket", an anti-parallel beta-sheet formed between Miro and its partner, as well as salt-bridges. The study uses AlphaFold2 to predict the interfaces and then test and verifies them using mutational analysis and cell biological assays or imaging. The Miro1 clients characterised in this way include previously known interactors such as CENPF, and interactors identified newly in this study such as MTFR1/2. It also addresses conservation of the interaction between human and yeast by characterising the interface between the yeast Miro1 ortholog Gem1 and Mdm34. Overall, the study suggests that the interaction is widespread, has probably evolved multiple times and is thus likely a versatile mechanism for the recruitment of cytosolic proteins to the mitochondrial surface. This is a carefully designed and carried out study that pinpoints a new molecular mechanism of organelle localisation. It will certainly inform numerous ongoing and future studies and therefore certainly be influential. Here are a few

suggestions for the authors on how to improve the manuscript:

- Fig 1B: A Miro1 construct which contains the GTPase1 and both ELM1 and 2 domains is indicated as binding and not-binding CENPF (+/-), while the overall experiment conclusion is that GTPase 1 and ELM1 are sufficient and necessary for binding. Can the authors comment on the result regarding this particular construct?

- The authors refer to the binding interfaces as "conformation", e.g. in line 43/44 page 2: "... bind Miro with a similar conformation", or page 5 line 35. While I agree that some structural elements of the interactions are clearly conserved (a major finding of the paper) I am not sure "conformation" is the right word here, since the regions are predicted to be unstructured hence the clients might have very different conformations in the domains bound to Miro even though the mode of binding is similar. Also, the conservation of these client regions is analysed / shown separately for the different clients, but it would be great if it could be compared also between clients - as this is important for identifying the binding site of VPS13D, and for the section in the Discussion that discusses generality of the Miro-binding structures (page 6, second paragraph).

- I don't find the phrasing of the concluding (final) sentence of the second Results section entirely convincing in the context of the results. Since Trak proteins do not require Miro to bind mitochondria, there is no evidence shown that their association with Miro as tested here is physiologically relevant.

- Line 39 on page 3: The search for the ELF-binding motif of CENPF yielded 5 candidates, and the focus was put on MTFR2. What were the other 4? Why MTFR2 was selected? Were the paralogs MTFR1 and 1L among the 5 hits? Since the conclusion of this section is that the motif search identified novel clients of the Miro-ELF pocket, at least discussing the other four hits seems important.

- Line 19/20 on page 5: Why does the difference in molecular weight matter? If I understand correctly, the authors want to emphasize that the difference between wt and mutant VPS13D is very small as it is a very large protein, but this seems irrelevant in the light of how point mutations work by specifically disrupting molecular interactions.

- In the first Results section, end of line 27 page 2 it is a little difficult to follow which residue are in Miro1 and which in CENPF. Perhaps add: CENPF-F2989 - a key phenylalanine...

Dear editor, dear William

Thank you so much for handling our manuscript. We were thrilled that the reviewers were in general positive about our study finding it “a good manuscript appropriate for publication in EMBO” (Reviewer 1), providing a “significant headway in understanding how Miro can bind multiple partners” (Reviewer 2), which “will certainly inform numerous ongoing and future studies and therefore certainly be influential” (Reviewer 3). We have improved the manuscript by taking their points into consideration and addressing them with new data and clarifications in the text. We have changed Figure 5C as per Reviewer #1’s suggestion, and added an analysis of Miro degradation as per reviewer #1 and #2’s suggestions. Several adaptations in the text have been made in response to all three reviewers. They are highlighted in red in the revised version. We wish to thank all three reviewers wholeheartedly for the time and effort they put into reading our work. See below.

Sincerely,

Christian Covill-Cooke
Benoît Kornmann

Referee #1:

The yeast two hybrid assay shows that CENPF D2991R impairs recruitment by Miro (Fig. 1F). Does this mutant also impair recruitment by Miro when expressed in mammalian cells?

This is an interesting question but its answer is difficult to get. Unlike the CENPF-F2989A mutation, the CENPF D2991R impairs, without abrogating the binding. It therefore requires a quantitative assay. Yeast-2-hybrid allows this quantitativity. In mammalian cells, however, we have two assays to assess CENP-F recruitment; the easy assay is to co-overexpress Miro and CENPF (or mutant thereof) by transient transfection. This assay isn’t quantitative as Miro overexpression might recruit CENPF regardless, with increased expression compensating for decreased binding. The better assay is to engineer the mutation in the CENPF locus directly (as we have done for the CENPF-F2989A, Peterka et al. PloS Genet 2019). In this assay, neither Miro nor CENPF are overexpressed and the effect of the mutation is therefore revealed in a physiological context. However, we hope that the reviewer will understand that this represents a significant endeavour for limited insight. Indeed, we have already used the CENPF-F2989A in vivo (with the proper assay), and the neighbouring CENPF D2991R is only additionally useful in the context of the charge swap assay, which we performed in the yeast 2 hybrid system.

Are there functional consequences for the mutations identified in this work? For example, does the mutation of CENPF (Fig. 1F) impair the reported redistribution of mitochondria by Miro and CENP-F to the cell periphery (Kanfer et al., Nat Comm 2015) or does the mutation of Parkin (Fig. 5H) prevent degradation of Miro after mitochondrial damage (Wang et al., Cell 2011)?

This is a vast and fascinating question that goes far beyond this manuscript. For instance, we have shown that engineering the CENPF-F2989A mutation in U2OS cells causes a similar mitochondrial spreading defect as lacking Miro or CENPF altogether (Peterka et al. PloS Genet 2019). However, mice carrying these mutations were healthy and fertile, indicating that the function of Miro-CENPF interaction was likely either in a specialised cell type or in special conditions, and that it was not necessary for development. Similarly, mutations that disrupt Trak-Miro or Myo19-Miro interactions would be fascinating to study. But studying their functional consequences would require a physiological setup, i.e. an animal model (as most of these interactions are metazoan, or even mammalian specific). This is beyond the scope of this paper. We have nevertheless assessed

mitophagy when Parkin bears the L119A mutation disrupting the Miro interaction. This interaction has been suggested to play important roles in Parkin biology. We found surprisingly, however, that the kinetics of Miro degradation, Parkin recruitment and auto-ubiquitylation were comparable to wild type Parkin (Fig. EV5C-D). This indicates that the conserved and well-documented interaction between Miro and Parkin is not strictly required for Parkin activity, as previously documented in Miro knockout/knockdown studies (e.g. Safiulina et al. EMBOJ 2019), or on its activity on Miro, but is likely involved in more subtle regulatory processes. Uncovering these processes would likely require a much more physiological setup than HeLa cells overexpressing Parkin and treated with uncouplers. This is, the reviewer will agree, outside the message of this paper.

The quality of Fig. 5C is not optimal and does not convey the message that it is supposed to convey: reduced recruitment to mitochondria of VPS13D^ΔGFP L2554A. This figure should be replaced with a better figure

We agree and have replaced it with a better figure.

Referee #2:

Referee #2 didn't have specific comments but general observations. We note the many strengths they found to our study. Below we address the weaknesses they identify.

Weaknesses:

The motivation of this study was to understand how Miro GTPases may be able to bind so many reported "clients", whether this was a competitive type of event, or if larger complexes may form, etc. The weakness then comes in the lack of biochemical confirmation of these interactions, and how they may actually compete functionally. The recruitment of cytosolic proteins (CENPF, Trak2, MYO19) to mitochondria may be mildly affected in Miro overexpression studies, but this is not proof that the effects are indeed due to the diminished Miro interactions.

We disagree that diminished recruitment is “no [evidence] that the effect are due to diminished Miro interaction” (as experimental scientists, we might want to leave the term “proof” to mathematicians). Interaction has been measured by Yeast 2 hybrid assay for CENPF, Trak1, Trak2, Myo19, MTFR1, MTFR2 and MTFR11. They match exactly what has been shown in the mitochondrial recruitment assays for CENPF (Peterka et al. PloS Genetics 2019), Myo19 and MTFR11. Note that we do not claim that Trak2 mitochondrial recruitment is affected, and we apologise if the reviewer has been led to assume this. Indeed, it is well known that Trak2 does not require Miro to localise to mitochondria. We therefore really struggle to interpret our data otherwise than the mutations impairing interaction with Miro.

There are no functional assays to determine the effects of these processes on mitochondrial motility, or mitophagy, or whichever additional functions they envision.

See response to reviewer 1, point 2. The present manuscript describes the molecular architecture of interactions, which, with the exception of MTFR1, 2 and 1L, are well-known in the literature, and for which functions have been proposed since two decades.

Our manuscript is not about the function, but about the molecular architecture of these interactions. While these well-documented functions (axonal mitochondrial transport, mitophagy, interorganelle lipid transport) are different for each client, they are operated by a similar binding architecture. Our manuscript is on the common aspects of Miro biology rather than on the divergent ones.

For instance, drosophila mutated for Miro or Milton somewhat phenocopy each other, with a common defect in axonal mitochondrial transport. The model has therefore been that the interaction of both proteins is necessary for their function. We now have the tools to test this idea, but the reviewer will agree that this is beyond the scope of the present study which, again, is about the molecular architecture of these interactions. For mitophagy, please see our response to reviewers 1 point 2.

The effects on VAPB recruitment seem rather minor (Fig 5D) and it's not clear this is proof that the consequence is due exclusively to an inhibition of Miro binding.

We have to assume that the reviewer means VPS13d instead of VAPB, because we have not performed experiments involving VAPB. Miro-VPS13d physical interaction has been demonstrated previously by the lab of Pietro de Camilli, where they show that Miro overexpression recruits VPS13D to mitochondria, and KO of both Miro1 and 2 prevents VPS13d binding to mitochondria (Guillem-Samander, JCB 2021). They show that even with Miro overexpression, only a fraction of VPS13D is recruited to mitochondria. Therefore, the enrichment we observe, which is of a factor 2 on average, is likely not far from the mark. The mutant, by contrast averages close to 1, which means no enrichment at all, and is what would be expected if Miro binding was indeed abrogated. Thus, if the recruitment is not as spectacular as for other factors, it is likely due to intrinsic properties of Miro-VPS13d interaction. But this does not mean that the interaction is not meaningful, as shown in the paper from the de Camilli lab.

Perhaps most confusing is their assertion that this interacting face forms a meaningful, functional interaction with MTFR1/2/1L (also called MTP18). This is an established inner membrane protein with three transmembrane domains that was first studied in the mid-2000s. It is difficult to conceive how it could interact with a cytosol-facing domain on Miro. The Miro1/2 double KO cells see normal import of the MTFR1 and 2 proteins, therefore this interaction is not essential to ensure the biogenesis of these inner membrane proteins. On the other hand, loss Miro1/2 did affect MTFR1L import, so perhaps this variant is not in the inner membrane? But this was not tested or followed in any functional way. The overexpression shows the mutant MTFRs accumulate in cytosol, however this may reflect structural alterations that abrogate protein targeting and insertion, rather than reflect functional binding to Miros. The targeting sequence of multispinning inner membrane proteins is not always so obvious and could have been interrupted by these mutations. The authors didn't mention the native topology or mitochondrial location of MTFP proteins, which was confusing. Is the MTFR1L76A (and F residue equivalents in MTFR2 and 1L) upstream of the first transmembrane domain (?) in a rather disordered region? I'm really not clear what the authors think this interaction with Miro is for, does it have to do with mitochondrial motility, or import somehow? it just seems very out of context and preliminary.

We do not understand this comment and can only assume that the reviewer is confusing MTFR1 and MTFP1. MTFP1 is indeed an inner-membrane protein with three TM domains, which is also referred to as MTP18. MTFR1, 2 and 1L, by contrast are soluble proteins. While there are conflicting reports about their submitochondrial localisation, recent papers about MTFR11 argue convincingly for an outer membrane localization (for instance MTFR11 is a target of the cytosolic AMPK, Tilokani et al, Sci Adv 2022). We do not, at this stage, have an answer to the functional role of this interaction, and consider that this lies beyond the scope of this study. The molecular function of MTFRs themselves is in fact quite mysterious.

Again, this study is about the molecular architecture of Miro-client interaction. Discovering the fact that this architecture involves binding to a short unstructured peptide opens the way to the discovery of clients *de novo*, which is the endeavour we pursued in our study of MTFR1/2/1L. We indeed

show that Miro is necessary and sufficient to recruit MTFR11 to mitochondria. The role of this interaction (like the function of MTFR11 itself) remains to be discovered.

Referee #3:

- Fig 1B: A Miro1 construct which contains the GTPase1 and both ELM1 and 2 domains is indicated as binding and not-binding CENPF (+/-), while the overall experiment conclusion is that GTPase 1 and ELM1 are sufficient and necessary for binding. Can the authors comment on the result regarding this particular construct?

This is a fair and important comment and the explanation is straightforward. As you can see in the figure below, this particular fragment (fragment A) does not express nearly as well as the others. We amended the legend to figure S1 to include the following sentence: "Growth of the second fragment was less robust than that of the first and fourth fragments because the expression of this fragment was poor."

- BAIT: Cenp-f "E" (2819-3114)
 PREY: Miro1 fragments
 X: GTP1-EF1-EF2-GTP2 +
 A: GTP1-EF1-EF2-GTP2 +/-
 B: ~~GTP1-EF1-EF2-GTP2~~ -
 C: ~~GTP1-EF1-EF2-GTP2~~ -
 D: ~~GTP1-EF1-EF2-GTP2~~ -
 E: ~~GTP1-EF1-EF2-GTP2~~ -
 F: GTP1-EF1-EF2-GTP2 +
 G: ~~GTP1-EF1-EF2-GTP2~~ -
 H: ~~GTP1-EF1-EF2-GTP2~~ -
 I: ~~GTP1-EF1-EF2-GTP2~~ -

- The authors refer to the binding interfaces as "conformation", e.g. in line 43/44 page 2: "... bind Miro with a similar conformation", or page 5 line 35. While I agree that some structural elements of the interactions are clearly conserved (a major finding of the paper) I am not sure "conformation" is the right word here, since the regions are predicted to be unstructured hence the clients might have very different conformations in the domains bound to Miro even though the mode of binding is similar.

We agree that "conformation" is not the most appropriate term for a binding that shows great flexibility in shape. We have replaced these occurrences by better chosen words (oftentimes "configuration" does the deal).

Also, the conservation of these client regions is analysed / shown separately for the different clients, but it would be great if it could be compared also between clients - as this is important for identifying the binding site of VPS13D, and for the section in the Discussion that discusses generality of the Miro-binding structures (page 6, second paragraph).

We do not observe conservation of the binding motifs across clients. Thus far, the only common residues are the F/L residues, often an acidic residue, the position of which can be on +2, +5 or +6 position (probably facing Miro's arginines periodically around the partially alpha-helical backbone), and often a Glycine residue at the -1 position. These similarities are too scarce to speak of conservation and it is instead more likely that these motifs have evolved independently several times in evolution. Rather than "conservation", it would therefore be more appropriate to speak of "convergence".

- I don't find the phrasing of the concluding (final) sentence of the second Results section entirely convincing in the context of the results. Since Trak proteins do not require Miro to bind mitochondria, there is no evidence shown that their association with Miro as tested here is physiologically relevant.

We understand this comment as follows: while we show that the discovered binding peptide is sufficient for interaction in yeast two hybrid, we haven't shown it was absolutely necessary in vivo, as other interfaces might also exist on Trak that bind Miro. This is a fair comment. Yet, the interaction between Miro and Trak has been first detected by Yeast 2 hybrid, using protein fragments that contain the same residues as we show bind Miro (Giot L et al. Science 2003). The interaction domain was further refined as a 225-residue fragment (McAskill et al. Mol Cell Neurosci 2009), which encompasses the peptide we discovered. All of this makes it likely that this peptide is both necessary and sufficient. Nonetheless, we agree with the reviewer that this is not formally demonstrated. We therefore changed the wording of the sentence as "Thus, while we cannot exclude at this stage that other molecular determinants play a role in the interactions, we find that the Trak proteins and MYO19 associate with Miro via a shared conserved binding pocket."

- Line 39 on page 3: The search for the ELF-binding motif of CENPF yielded 5 candidates, and the focus was put on MTFR2. What were the other 4? Why MTFR2 was selected? Were the paralogs MTFR1 and 1L among the 5 hits? Since the conclusion of this section is that the motif search identified novel clients of the Miro-ELF pocket, at least discussing the other four hits seems important.

Indeed, this search did not ambition to be either specific (only one of five hits were likely to bind Miro) or sensitive (none of the previously known Miro interactors matched the criteria). The other 4 proteins were DLST, NT5DC3, TOMM34 and SLC25A3. Of this list, only TOMM34 was on the outer membrane. Contrary to MTFR2, AlphaFold did not predict interaction between TOMM34 (nor any of the other identified candidates) and Miro. MTFR1 and MTFR11 were not in the list because they bear FADV and LADI motifs, respectively. Yet, with all its shortcomings, this search makes the point that new interactors for Miro can be found on the basis of their sequence. We have added a section in the discussion to discuss the shortcomings of our approach, which indeed forced us to strongly bias our selection by using mitocarta (a comment also made by reviewer 2).

- Line 19/20 on page 5: Why does the difference in molecular weight matter? If I understand correctly, the authors want to emphasize that the difference between wt and mutant VPS13D is very small as it is a very large protein, but this seems irrelevant in the light of how point mutations work by specifically disrupting molecular interactions.

Previous attempt at mapping Miro-Vps13D interface have been made by hacking away large portions of the protein (Guillen-Samander JCB 2021). The conservative L>A mutation makes much smaller and therefore much more specific alterations to the protein. This is what we tried to emphasize, but the reviewer is correct that this information on molecular weight is irrelevant. We have therefore removed it.

- In the first Results section, end of line 27 page 2 it is a little difficult to follow which residue are in Miro1 and which in CENPF. Perhaps add: CENPF-F2989 - a key phenylalanine...

That is a great idea which we have followed, and which increases the clarity of our paper.

Dear Benoît,

Thank you for submitting the revised version of your manuscript, which addresses the concerns of the referees. This revised version has now been re-reviewed; I attach the second referee reports to the bottom of this mail. As you will see, notwithstanding suggested points for additional discussion raised by Referee 2 (which I would like you carefully to consider), you have addressed the referees' concerns to their satisfaction.

Before I can formally accept your manuscript for publication, however, there are some remaining editorial points which need to be addressed. In this regard would you please:

- remove the main and EV figures from the manuscript text,
- remove the 'Funding' section of the manuscript and include funding details in the 'Acknowledgements' section,
- alphabetize the reference list, using et al. after the tenth author name,
- remove the DOI number from the reference list,
- rename the Conflict of Interest statement the 'Disclosure and Competing Interests' statement,
- remove the author credit section from the manuscript,
- the data which are uploaded as a Zip folder need to be renamed as Dataset EV1 (the callout in the ms need updating too),
- include a table of contents with page numbers, renaming it Appendix Figure S1 (the callout in the ms is OK); the legend for the dataset needs to be removed from the file and provided together with the dataset,
- provide a scale bar and its definition for figure EV5d,
- use the following order for the sections of your manuscript: Title page, Abstract & Keywords, Introduction, Results, Discussion, Materials & Methods, Data Availability, Acknowledgments, Disclosure Statement & Competing Interests, References, Figure Legends, Tables with legends and finally Expanded View Figure Legends,
- rename Expanded View figure files so that Supp. Fig. 1 is called Figure EV1, etc.,
- name the tables on pages 9 and 10 Table 1 and Table 2, and
- The EMBO Journal now provides figure legends in a run-on style; please adapt you legends accordingly.

EMBO Press is an editorially independent publishing platform for the development of EMBO scientific publications.

Best wishes,

William

William Teale, PhD
Editor
The EMBO Journal
w.teale@embojournal.org

Please remember: Digital image enhancement is acceptable practice, as long as it accurately represents the original data and

conforms to community standards. If a figure has been subjected to significant electronic manipulation, this must be noted in the figure legend or in the 'Materials and Methods' section. The editors reserve the right to request original versions of figures and the original images that were used to assemble the figure.

We realize that it is difficult to revise to a specific deadline. In the interest of protecting the conceptual advance provided by the work, we recommend a revision within 3 months (11th Mar 2024). Please discuss the revision progress ahead of this time with the editor if you require more time to complete the revisions. Use the link below to submit your revision:

Referee #1:

I consider this excellent revised manuscript acceptable for publication

Referee #2:

I thank the authors for responding to my concerns, and I apologize for confusing MTFP1 with MTFR1, and mistyping VapB instead of VPS13D in my review. The latter was a mistyping, I did understand the work was with VPS13D, and I am aware of the DiCamilli work. I understand my review must have been very frustrating for the authors, and again, I apologize. I do, however, remain a bit confused on the localization of MTFR1 and MTFR2, while MTFR1I seems to be a true OMM protein. Given that the interaction of these proteins with MIRO is a very novel aspect of the study, it would be appreciated if a few more sentences might be included explaining what these proteins are linked to. The recent paper cited described MTFR1I as an AMPK substrate regulating mitochondrial dynamics, and MTFR2 is linked to many cancer studies. Since was in such error in my review, I looked more carefully on Uniprot links and see that MTFR1 and MTFR2 were included as baits in the study by Gingras/Shoubridge where they did bioID on ~100 mitochondrial proteins. The hits for both MTFR1/2 were almost exclusively outer membrane (TOMM20, RHOT1 (replicating your work here), MTCH2, SCL25A46, etc). I think it might be helpful to include this reference PubMed ID: 32877691 as support for the OMM localization of all 3 proteins.

My main concern was the limited functional data for the interactions reported in the study. I was very supportive of the in silico approaches using alphaFold, and I understand the use of the Y2H and overexpression/recruitment studies. I do believe these approaches are somewhat limiting and it's difficult to infer physiology from them. I still don't really understand how these proteins would compete for binding to MIROs to exert specific functions. However, since the author and both other reviewers consider functional studies to be well beyond this work, I defer. What is presented is of high quality and we will wait to learn what it all means in later work from the lab.

Referee #3:

In this revised manuscript and in the rebuttal letter, the authors have addressed all my concerns adequately. This paper on how Miro binds to its clients should now be published in EMBO Journal. Congratulations to the authors on this important work!

Dear editor, dear William

Thank you for handling our manuscript. We are grateful to all three reviewers for their positive comments. As Reviewers #1 and #3 do not include any comments that require addressing, we have only included our response to Reviewer #2 below. We would like to thank all three reviewers for their time and valuable input on this manuscript.

Yours sincerely,

Christian Covill-Cooke

Benoît Kornmann

Referee #2:

I thank the authors for responding to my concerns, and I apologize for confusing MTFP1 with MTFR1, and mistyping VapB instead of VPS13D in my review. The latter was a mistyping, I did understand the work was with VPS13D, and I am aware of the DiCamilli work. I understand my review must have been very frustrating for the authors, and again, I apologize. I do, however, remain a bit confused on the localization of MTFR1 and MTFR2, while MTFR11 seems to be a true OMM protein. Given that the interaction of these proteins with MIRO is a very novel aspect of the study, it would be appreciated if a few more sentences might be included explaining what these proteins are linked to. The recent paper cited described MTFR11 as an AMPK substrate regulating mitochondrial dynamics, and MTFR2 is linked to many cancer studies. Since was in such error in my review, I looked more carefully on Uniprot links and see that MTFR1 and MTFR2 were included as baits in the study by Gingras/Shoubridge where they did bioID on ~100 mitochondrial proteins. The hits for both MTFR1/2 were almost exclusively outer membrane (TOMM20, RHOT1 (replicating your work here), MTCH2, SCL25A46, etc). I think it might be helpful to include this reference PubMed ID: 32877691 as support for the OMM localization of all 3 proteins.

My main concern was the limited functional data for the interactions reported in the study. I was very supportive of the in silico approaches using alphaFold, and I understand the use of the Y2H and overexpression/recruitment studies. I do believe these approaches are somewhat limiting and it's difficult to infer physiology from them. I still don't really understand how these proteins would compete for binding to MIROs to exert specific functions. However, since the author and both other reviewers consider functional studies to be well beyond this work, I defer. What is presented is of high quality and we will wait to learn what it all means in later work from the lab.

We thank the reviewer for these comments and have now included the suggested reference as support for the sentence “MTFR1, MTFR2 and MTFR1L localize to mitochondria”. This study does not, in fact, use any of the MTFRs as bait, but detects them as prey in some of the experiments. The reviewer is perhaps confused due to a typo in the study where the bait MTRF1L (mitochondrial release factor 1-Like) is mistyped in the supplementary table 2 as MTFR1L. The authors have been informed of this typo and are taking appropriate action.

Whilst this paper does support a OMM localisation for all three MTFRs, and there are indeed other papers studying these proteins, we are hesitant to add too many further details, including statements on the submitochondrial localization, as conflicting evidence is present in the literature, and as the work on all three proteins is far from extensive. We, therefore, believe that what we have included is an honest representation of the work that is relevant to the manuscript: namely, that they localise to mitochondria and that they have a role in regulating mitochondrial morphology.

Dear Benoît,

I am pleased to inform you that your manuscript has been accepted for publication in the EMBO Journal.

Congratulations! I am really glad this all worked out and sure the article will be well-appreciated.

Best wishes,

William

William Teale, PhD
Editor
The EMBO Journal
w.teale@embojournal.org
